# Lunar rock investigation and tri-aspect characterization of lunar farside regolith by a digital twin

Liang Ding [1,9] ✉, Ruyi Zhou [1,9], Tianyi Yu [2,9], Huaiguang Yang [1], Ximing He [2], Haibo Gao [1] ✉, Juntao Wang [3,4], Ye Yuan [1], Jia Wang [2], Zhengyin Wang [1], Huanan Qi [1], Jian Li [2] ✉, Wenhao Feng [1], Xin Li [2], Chuankai Liu [2,5], Shaojin Han [2], Xiaojia Zeng [3,4], Yu-Yan Sara Zhao [6], Guangjun Liu [7], Wenhui Wan [8], Yuedong Zhang [2], Saijin Wang [2], Lichun Li [2], Zongquan Deng [1], Jianzhong Liu [3,4] ✉, Guolin Hu [2], Rui Zhao [2] & Kuan Zhang [2]

Yutu-2 rover conducted an exciting expedition on the 41st lunar day to investigate a fin-shaped rock at Longji site (45.44°S, 177.56°E) by extending its locomotion margin on perilous peaks. The varied locomotion encountered, especially multi-form wheel slippage, during the journey to the target rock, established unique conditions for a fin-grained lunar regolith analysis regarding bearing, shear and lateral properties based on terramechanics. Here, we show a tri-aspect characterization of lunar regolith and infer the rock's origin using a digital twin. We estimate internal friction angle within 21.5°–42.0° and associated cohesion of 520-3154 Pa in the Chang'E-4 operational site. These findings suggest shear characteristics similar to Apollo 12 mission samples but notably higher cohesion compared to regolith investigated on most nearside lunar missions. We estimate external friction angle in lateral properties to be within 8.3°–16.5°, which fills the gaps of the lateral property estimation of the lunar farside regolith and serves as a foundational parameter for subsequent engineering verifications. Our in-situ spectral investigations of the target rock unveil its composition of iron/magnesium-rich low-calcium pyroxene, linking it to the Zhinyu crater (45.34°S, 176.15°E) ejecta. Our results indicate that the combination of in-situ measurements with robotics technology in planetary exploration reveal the possibility of additional source regions contributing to the local materials at the Chang'E-4 site, implying a more complicated geological history in the vicinity.

After more than half a century of lunar nearside exploration, recent scientific research on the Moon is focused more on earlier geological history, deeper internal structure, and far-reaching habitability, to unravel fundamentally important questions on longer timescales, further spatial distances, and of greater significance for human life.

Space probes are required on unexplored regions of great scientific value, such as the lunar farside[1] and the south pole[2], which are usually incidental to greater difficulty. As the oldest and largest impact basin on the Moon, the South Pole-Aitken (SPA) basin is one of the most appealing farside places that is supposed to have exposed the lunar

lower crust and probably upper mantle materials[3,4], and promising to reveal the indeterminate evolution of the early Moon with oldest mare basalts ever detected[5,6]. Therefore, targeting at the floor of the SPA basin, the Chang'E-4 (CE-4) mission[7] soft-landed at the eastern edge of the mare-containing Von Karman crater in 2019 (Fig. 1a), within the ejecta field of the nearby Finsen crater[3], and deployed a Yutu-2 rover to investigate the topographic and mineralogical composition, as well as the subsurface stratigraphy of the roving area[8].

Estimates of lunar crustal thickness obtained from the Gravity Recovery and Interior Laboratory (GRAIL) mission[9] corroborate the notion that the SPA impact event likely excavated materials deep into the mantle[10], and there presents a large excess of mass in the lunar mantle under the SPA[11]. Additional evidence from remote sensing, impact modeling, and geological analyses indicates that the SPA impact-ejected ilmenite-bearing cumulates (IBCs) and potassium, rare-earth elements, and phosphorus (KREEP)-bearing rocks from the uppermost mantle[12,13]. Continuous spectral reflectance data acquired by the Spectral Profiler instrument aboard the lunar explorer Selenological and Engineering Explorer (SELENE)/Kaguya reveal enriched FeO contents in the central depression of the SPA, indicating the presence of mafic materials such as impact melt breccia[14]. Orbital spectral observations of the materials within the SPA strongly suggest the excavations of the lunar mantle; however, low-Ca pyroxene (LCP)-rich rocks are more numerous and more widely distributed than olivine-rich rocks,

dominating spectral signatures of the mantle-derived SPA impact melts[4,15–19]. Whether the sparse distribution of olivine-rich materials within the SPA is due to the lack of in-situ measurements with high resolution, or is it indicative of the layered structure of the lunar mantle, a horizontal heterogeneity in mantle composition, or the impact origin of the basin, remains open questions. Notably, the definitive identification of mantle materials, whether on the lunar surface or in the analysis of returned samples, remains elusive.

The properties of lunar regolith contains critical information about the nature and evolution of the Moon and surrounding space environment, and are of significant importance in engineering considerations. While numerous robotic and manned missions have conducted in-situ experiments to understand the physical and mechanical properties of lunar regolith[20,21], these efforts have primarily covered a limited portion of the lunar surface on the nearside, with no detailed properties of lunar regolith on the Moon's farside being available prior to the Chang'E-4 mission. The absence of specific payloads or various locomotion state measurements required for identification has left the mechanical properties of farside lunar regolith, especially longitudinal and lateral properties, poorly understood.

As the first successful mission launched within the vicinity of the SPA basin, Chang'E-4 provides a unique opportunity for mantle-derived material investigation and in-situ measurements of the lunar regolith at the farside of the Moon. During Yutu-2's expedition of the

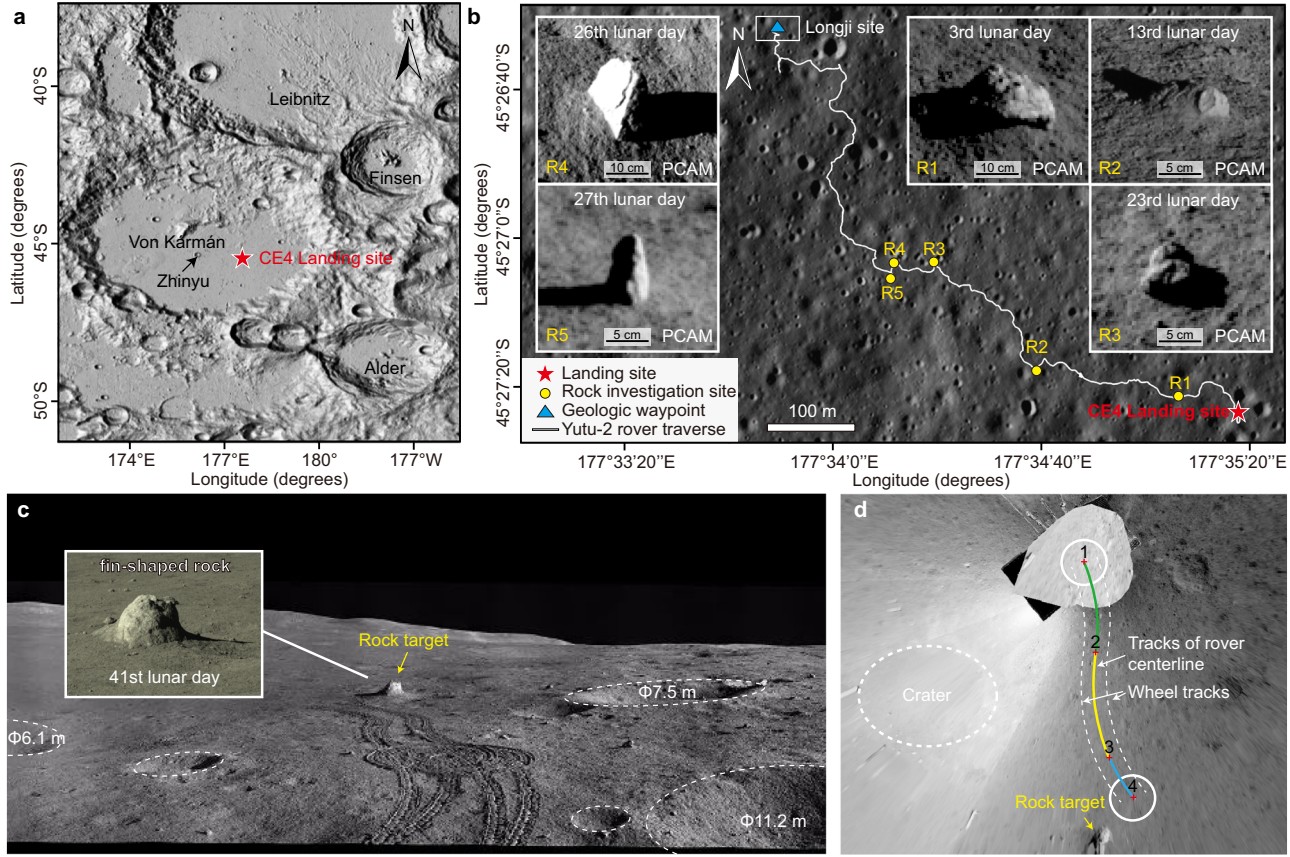

**Fig. 1 | Geomorphological context of the Chang'E-4 landing site at different scales. a** The shaded relief of the region around the Von Kármán crater derived from the Lunar Reconnaissance Orbiter (LRO) + Laguya merged topography products. **b** Yutu-2 rover's traverse map and rock investigation sites during the first 41 lunar days. The base map is a high-resolution (0.9 m per pixel) digital orthophoto map (DOM) obtained by the Lunar Reconnaissance Orbiter Camera (LROC) Narrow Angle Camera (NAC) (M1303619844 L/R and M1303640934 L/R). These insets are the associated rock targets shown in Pancam images. **c** Local landform of the Longji site seen in the panoramic image taken at the dormant point of the 41st lunar day.

The inset was taken by the Pancam at the forenoon of the 41st lunar day before the investigation. **d** The planned path of Yutu-2 to approach the exposed rock. The path consists of three curved movement sections divided by points 1–4, and their lengths (curvature) are 3.673 m (0.154 m⁻¹, green curve starting from point 1 to point 2), 4.282 m (−0.132 m⁻¹, yellow curve starting from point 2 to point 3), and 1.852 m (−0.138 m⁻¹, blue curve starting from point 3 to point 4). The rock target in subfigure d is the same one as labeled in subfigure (**c**). The crater outlined by a dotted circle in subfigure (**d**) is the one labeled with a 7.5 m diameter in subfigure (**c**).

first 41 lunar days, it has been navigating towards the northwest (Fig. 1b) for 1142.39 m until 8 April 2022, and investigated several exposed rock fragments[22–29] along the route. An olivine-norite rock was detected on the 3rd lunar day, and shed light on the composition of the lunar interior and the lunar magma ocean (LMO) crystallization[24]. More ancient persevered materials are expected to be found to enrich our understanding of the composition, formation, and subsequent evolution of the lunar crust and mantle[30]. On the 41st lunar day, Yutu-2 observed a fin-shaped rock of scientific interest but located over complex terrains, which probably present major slipping and skidding challenges to rover's mobility. Digital twin[31], built upon the expert knowledge and real data collected from the physical system, facilitates a more precise simulation across different temporal and spatial scales, allowing for detailed planning and analysis of the rover's movements to deal with these challenges. The wheel-regolith interaction, refined through the exchange of data between the virtual and physical systems in digital twin, also provides a promising way to reveal unknown regolith properties at the lunar farside.

In this work, we present a slip/skid-risky but successful venture of Yutu-2 rover to a peculiar fin-shaped rock enabled by digital twin with associated spectral investigation results, and the tri-aspect property identification of the farside lunar regolith achieved on special slipping and skidding states.

## Results

### Topographic and mobility hazards analysis

The target rock was located at Longji site (45.44°S, 177.56°E), which was surrounded by multiple meter-level-sized craters and at an elevation of −5922.06 m. In contrast with the surrounding potholed surface, the rock was abruptly standing on the surface and looked like the fin on a dragon's back as displayed in a panoramic image (Fig. 1c). From its surface morphology, we inferred that it was likely sputtered from other places after impacts and promising to be lunar lower crust or upper mantle materials. As shown on the contour plot of the local topography (Supplementary Fig. 1a) generated by photogrammetry[32], the rock was on the south side of the rover at a straight-line distance of nearly 11 m, and there were two craters of 6.1 m and 11.2 m in diameter ahead of the rover on the left and right sides respectively, only setting aside a narrow uneven passage doubtful in accessibility. The slope of the local confined route was calculated to be around 8.86° at maximum with a 5.38° slope at the end, and most large slopes were distributed at impact crater rims (Supplementary Fig. 1b). To approach the distinctive rock through this wandering footpath, Yutu-2 rover was not only required to move across the steep slope, but also to reach a comparatively gentle platform located at the inner crater wall for its subsequent close-range observation and visible and near-infrared imaging spectrometer (VNIS) detection. Besides, small-sized gravels protruding from the surface on this route (Supplementary Fig. 1c) further complicated the traversal.

Such robotic geological exploration to desired scientific interesting target relies significantly on the successful traversal of rovers across diverse extraterrestrial terrains. However, from the locomotion perspective, this potential traverse was full of mobility hazards and uncertainties. On the one hand, the Yutu-2 rover was likely to suffer wheel skidding[33] of high risk when moving downwards on such a slope. Furthermore, uncontrollable lateral slippage would inevitably occur with broken nonholonomic constraints of the rover during the traverse, thus the path-following accuracy of Yutu-2 is hard to guarantee. Locomotion failures on harsh or slippery terrains have ever brought serious consequences, such as reduced tractive performance, deviation from planned trajectories, and in the worst-case scenario, becoming immobilized and permanently trapped, exemplified on previous missions on the Moon (like Lunar Roving Vehicle (LRV)[34], Luna 21 and Lunokhod 2[35]) and Mars exploration missions[36,37]. In addition, the nearest dormancy-available zone was in the northwest which was completely opposite to the forward direction to the target. To fulfill the strict requirements of the rock investigation and the subsequent wake-up, the rover had to move efficiently in a round-trip (returning to the departure point) within the same lunar day and collect high-quality data in between.

Considering the traversability (determined by slope steepness, terrain roughness, step height, and other factors; Supplementary Fig. 1d), an S-shaped path of 9.8 m for the onward journey (Fig. 1d) was planned[38], consisting of three movement sections with a regulable short approaching section in a straight line (Supplementary Table 1). To keep the mission continue, movements on harsh terrains with slipping risk must be justified to remain within the acceptable bounds of the rover's mobility safety margin prior to execution. However, predicting rover mobility is a complex task affected by surface material properties, terrain geometry, mobility system configuration, and it is especially challenging on soft terrains[39] due to the complexity of the wheel-terrain interaction. It's difficult to assess the safety of the planned onward traverses, further compounded by the accumulation of substantial uncertainties during the mission.

### Digital twin-based mobility evaluation

To address these issues mentioned above, a digital twin system[31] (Fig. 2) was developed to comprehensively evaluate the mobility performance of the Yutu-2 rover along the planned route. A virtual lunar rover (Supplementary Fig. 2a) of the same configuration as the Yutu-2 (Supplementary Fig. 2b) was generated with dynamic models (Supplementary Fig. 2c), and a local terrain model with geometric and physical properties[40] (see Methods subsection rover locomotion simulation) as well as realistic textures was reconstructed in the digital space based on transmitted images[41]. Both the virtual rover and the Yutu-2 shared identical control commands to maintain a consistent pace in different spaces, and the data collected by the Yutu-2 rover on the Moon were transferred to the Earth to update the digital world close to its physical counterpart. The interaction of the action and data in between bridged the two spaces, and promoted them to evolve synchronously. A series of numerical calculations (prediction, analytics, optimization, and evaluation) were conducted to assess the safety and efficiency of the planned locomotion. In addition, the data (rover states and images) collected during the performed movement were in turn used for further refined terrain property estimation, rectifying the digital models towards higher fidelity.

To simulate the behavior of a rover on the lunar surface, a wheel-terrain interaction model with consideration of multi-form wheel slippage[42–44] on soft and deformable terrain (Supplementary Fig. 3) was developed in the digital twin system. Based on the dynamics and terramechanics, Yutu-2's locomotion was predicted in iterative loops[40] (Supplementary Fig. 4). A wide range of parameters for regolith was set in the wheel-terrain interaction model to cover most cases, e.g., the internal friction angle $\varphi$ varied from 25° to 55°, and the lateral shearing deformation modulus $K_y$ (the soil deformation modulus in lateral direction[44]) was within the range of 15–45 mm. Predictions of the outbound traverse on various parameters generated a cluster of curves, which showed the slippage severity under general circumstances. With the extension of routes, all predicted paths deviate from the planned path in the longitudinal and lateral directions due to the slippage, and end within the dotted circle that is not beyond 0.762 m from the planned destination (Fig. 3a–d). Despite the lack of determined terrain parameters, the locomotion of the rover follows specific laws regulated by terrainmechanic principles as shown in Fig. 3a–d. During the outbound journey, simulation results (Supplementary Fig. 5) show that rover wheels work mostly in skid condition (a negative slip condition when the circumference velocity of wheel is smaller than the traveling velocity of wheel), and suffer lateral slippage to a certain extent. The relative simulation errors for slip ratio prediction, as verified in prior ground experiments, consistently remained less than

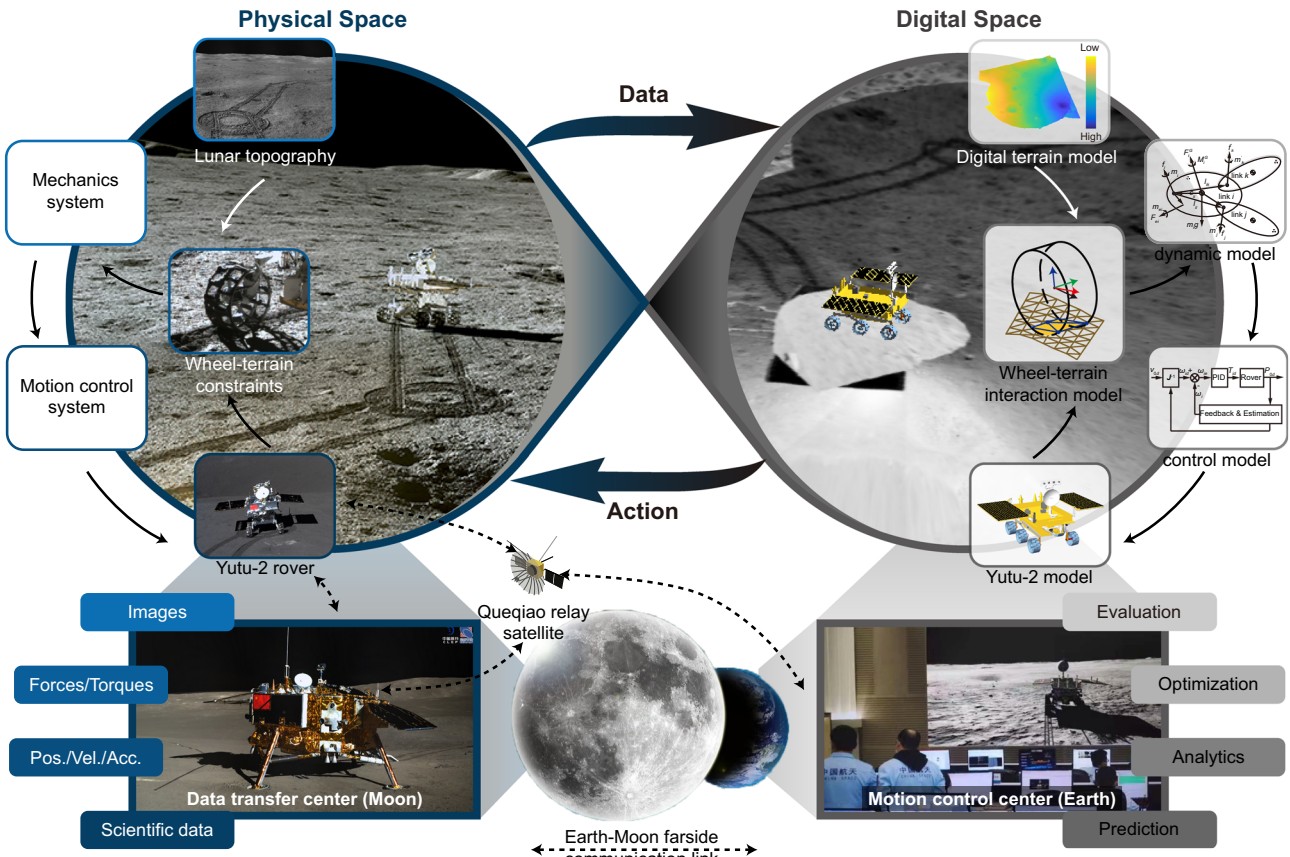

**Fig. 2 | Digital twin system for Yutu-2 rover's locomotion analysis.** The digital twin system seamlessly integrates physical and digital spaces, with interconnected data and actions. The data transferred from the physical space to the digital space includes images, locomotion information of the rover, scientific data, etc. The locomotion commands planned and evaluated in the digital space are taken as the action and sent to the physical space and carried out by the Yutu-2 rover. The digital space mirrors the corresponding physical system, encompassing terrain, rover, and physical constraints. The virtual and Yutu-2 rover share identical control commands for consistent pacing across spaces, and the Moon-collected data is transmitted to Earth, rectifying the digital world towards higher fidelity. Numerical calculations (prediction, analytics, optimization, and evaluation) on the motion control center on Earth generate actions, while collected data (scientific data, positions/velocities/accelerations, forces/torques, and images) centralized on the Moon's data transfer center, is transferred to Earth via the Queqiao relay satellite. The dotted lines represent communication links. The source file of the Yutu-2 rover is from: https://rb.gy/7lloxg. The source file of the wheel-terrain constraints image is from: https://rb.gy/l9gb2f. The source file of the background image of physical space is from: https://rb.gy/yu7too. The source file of the data transfer center (Moon) image is from: https://rb.gy/9adfze.

3.4%[40]. For lunar regolith with larger internal friction angle $\varphi$, its shear strength (the ability to resist relative skewing between particles) is stronger, resulting in the increase of the rover wheel slip ratio (whose median increases from −0.035 to −0.002 as the internal friction angle $\varphi$ increases from 25° to 55°, Supplementary Fig. 6a). For lateral locomotion, when the lateral deformation modulus $K_y$ of the lunar regolith in the wheel-terrain interaction model increases, the lateral shear effect between the regolith and the wheel is weakened. As a result, the lateral skid of the wheel becomes more severe, as represented in the increase of the side slip angle $\beta$ along the route (from 0.742° to 1.635° of the median, Supplementary Fig. 6b). The lateral model employed for representing wheel side force underwent rigorous validation to enhance its predictive performance for steering trajectories, achieving a final state error of less than 15%[43]. Additionally, this model exhibits a high degree of accuracy in estimating the rover's orientation[43].

Against the mission-defined threshold, these onwards traverses were also quantitatively evaluated in the endpoint position, the wheel slip ratio, and the side slip angle, which were deemed within the safety margin, and successive movements were carried out. As a result, the arrival position of Yutu-2 came out 0.367 m away from the planned destination as measured, and close to the center among the predicted termination scatters. The small discrepancy between the predicted state and the real-world state underscores the robustness and efficacy of the digital twin system, uniquely positioned to enhance decision support and elevate the mission success rate.

**Mechanical property identification of the farside lunar regolith**
The locomotion performed by Yutu-2 rover was further used to identify the mechanical parameters of the local regolith, which also rectified terrain models in the digital world. The regolith of the CE-4 site was mainly produced by in-situ weathering of the local basalt that erupted 390 million years ago[27,45], although a certain amount of impact-ejected materials had contributed to the region from Finsen, Zhinyu, and other craters. Observations from the orbit suggested that the CE-4 surface materials contain about 13 wt% FeO, 1.5 wt% $TiO_2$, and <2 ppm Th, which implied that the regional regolith was basaltic, low in titanium, and low in Procellarum KREEP Terrane (PKT) material (i.e., Lunar Procellarum KREEP Terrane)[46]. Additionally, some forsteritic olivine and magnesium-rich orthopyroxene materials had been detected in the regolith by the Yutu-2 rover[47]. The regolith particles were fine in size, and some were conglobated together sticking to the wheels after grinding, as shown in the Hazcam images (Supplementary Fig. 7), which preliminarily indicated its relatively strong cohesion in mechanical properties[48]. More fine-grained parameter identification was carried out on the basis of terramechanic principles with the simulation support on the digital twin system. Using the rover wheels as the test device, terrain mechanical

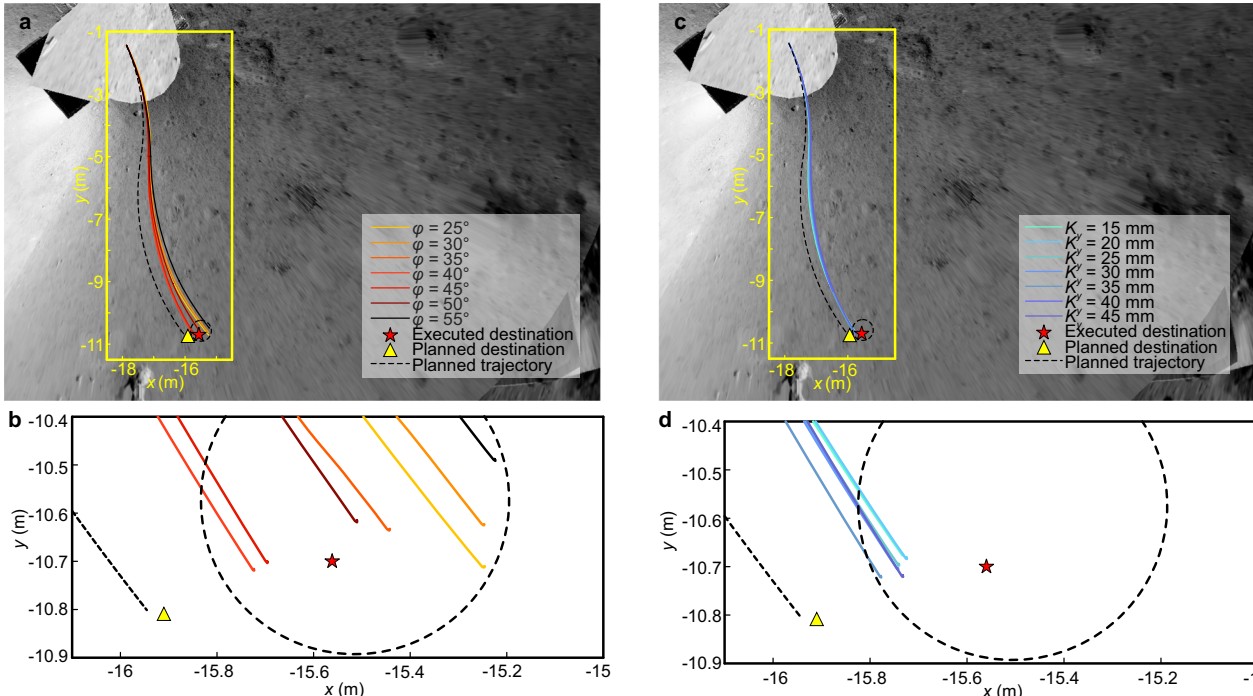

**Fig. 3 | Yutu-2's locomotion performance analysis based on a digital twin system. a** Yutu-2's onwards locomotion predictions under different internal friction angles $\varphi$ of the lunar regolith. **b** Enlarged view of Yutu-2's onwards locomotion predictions under different internal friction angle $\varphi$ of the lunar regolith around the executed destination. This subfigure shares the same legend as subfigure a. **c** Yutu-2's onwards locomotion predictions under different lateral shearing deformation modulus $K_y$ of the lunar regolith. **d** Enlarged view of Yutu-2's onwards locomotion predictions under different lateral shearing deformation modulus $K_y$ of the lunar regolith around the executed destination. This subfigure shares the same legend as subfigure (**c**). The terrain mechanical parameters in the simulation are listed in Supplementary Table 8.

parameters regarding bearing, shear, and lateral properties were estimated according to the terramechanics model for grouser wheels (see Methods subsection wheel-terrain interaction modeling).

The curves reflecting the regolith bearing characteristics under various wheel sinkage were predicted in comparison to typical soil samples on Earth (Fig. 4a) using models considering shear effects. The wheel sinkage $z$ of the traverse towards the rock was measured to be about 8 mm on average by reconstructing the wheel track surface with images, and no larger than 15 mm. When the sinkage exponent $N$ was assumed to be 1.0 (the value deduced from Apollo missions[49]), the equivalent stiffness $K_s$ varied from 556.7 to 1443 kPa m$^{-N}$ under an estimated sinkage $z$ of 8–15 mm. On other traverses, the wheel sinkage could be smaller but typically no less than 5 mm[48], and such condition corresponded to a smaller sinkage exponent $N$ around 0.87, because $K_s$ almost reached its upper bound with limited adjusting capability when the sinkage was small. Therefore, the bearing parameter of the farside regolith was bounded in the orange region ($N = 0.87$–1.0; $K_s = 556.7$–1443 kPa m$^{-N}$) shown in Fig. 4a, and compared with the terrestrial soil samples, it was analogous to that of the sandy loam measured in Michigan. When both the sinkage exponent and the equivalent stiffness modulus were supposed to be typical values of lunar regolith[49] ($N = 1.0$; $K_s = 820$ kPa m$^{-N}$), the corresponding wheel sinkage was about 12 mm, larger than the average sinkage of 8 mm measured in this mission, implying that the regolith at the Longji site was stronger in bearing strength than the counterpart of Apollo missions. With an average wheel sinkage of 8 mm along the rover's trace, the estimated bearing capacity reached approximately 4 kPa. This value was notably higher compared to the bearing capacity measured along the Lunohkod-1 traverse, where the uppermost layer at a depth of 1 cm exhibited a relatively smaller bearing capacity, falling within the range of 2–3 kPa[21]. This finding further substantiates the enhanced bearing strength of the lunar regolith at Longji site.

As for the shear property (specifically refer to the longitudinal shear property) of the farside regolith, no in-situ identification work has been carried out before the landing of the first lunar farside prospector, Yutu-2 rover. Nevertheless, it is still a challenging work for Yutu-2 because it has no specific payloads for that, and the interaction-based identification method requiring shear-related variables (such as the drawbar pull $F_{DP}$ or the longitudinal shear stress $\tau_x$ of the rover wheel) which cannot be measured directly onboard. Benefiting from the dynamic simulation on the digital twin system, the drawbar pull of rover wheels can be simulated and predicted instead, making plotting the shear characteristics curves under different wheel slip ratios possible, as shown in Fig. 4b. Since the rover mostly worked in skid condition (denoted by $s < 0$) and the wheel slip ratio $s$ was almost no less than −0.1 during the traverse, the shear parameters were bounded in between. Taking the internal friction angle (denoted by $\varphi$) of 42° and the cohesion (denoted by $c$) of 520 Pa (the typical values of the lunar regolith) as boundary conditions, then the shear parameters of the farside regolith were estimated to be within the orange bounded region ($\varphi = 21.5°$–42.0°, $c = 520$–3154 Pa). Compared with the lunar regolith properties identified at other nearside landing sites[49] (Fig. 4c), the longitudinal shear property (characterized by cohesion and internal friction angle) of the lunar regolith at Longji site was closest to that estimated in the Apollo 12 mission using direct shear method[49]. However, its cohesion was mostly larger than that measured at other sites, which was consistent with the great soil adhesion phenomenon observed on the wheel. The heightened degree of cohesion can be attributed to the increased presence of agglutinates within the lunar regolith, which constitute the principal product of space weathering[50]. Consequently, this may suggest that the local lunar soil is subject to longer periods of exposure and more pronounced space weathering effects, leading to an enhanced level of regolith maturity[50].

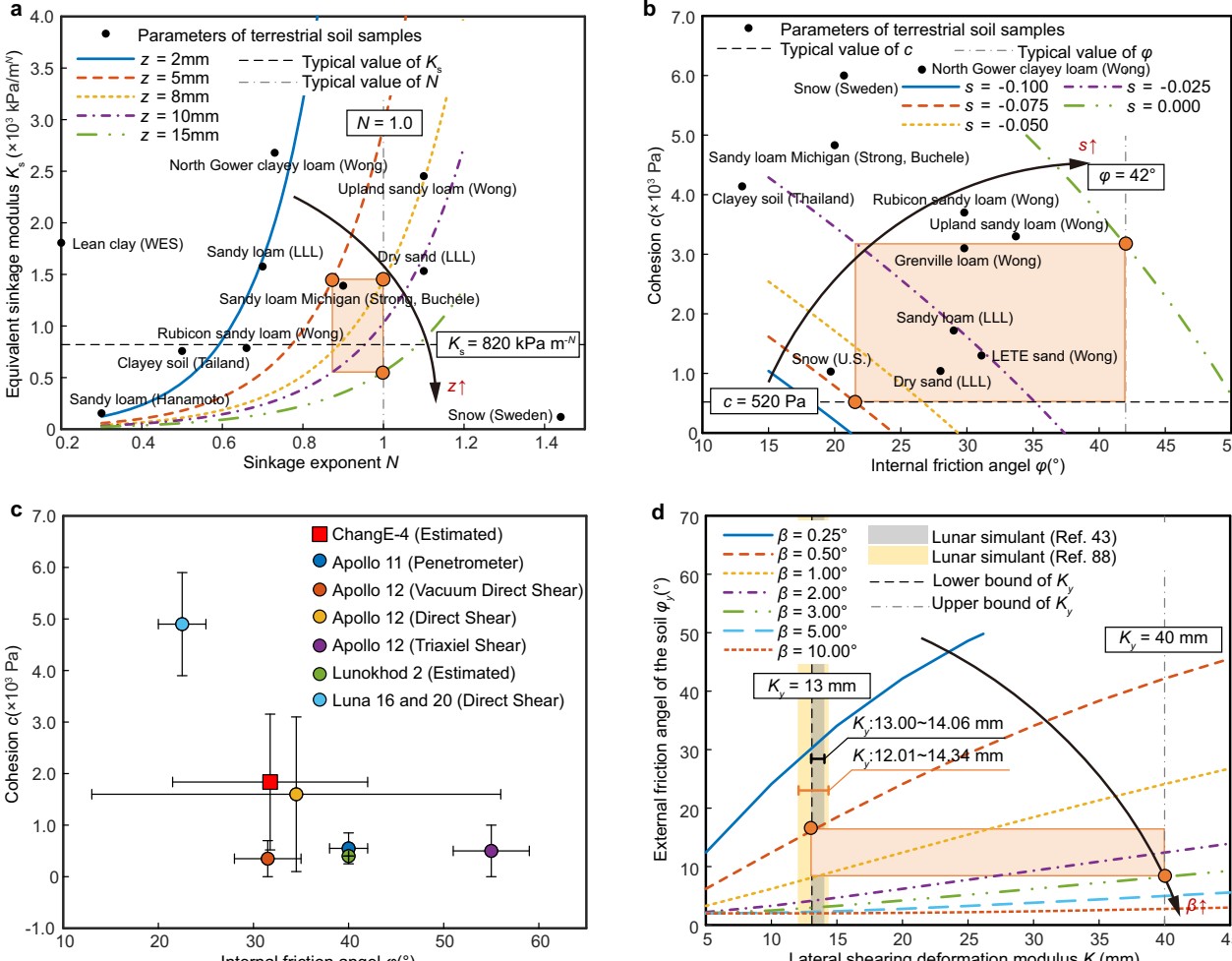

**Fig. 4 | Parameter analysis of the lunar regolith at the Longji site. a** Lunar regolith bearing parameters analysis in comparison to terrestrial soil samples. The bearing parameters of lunar regolith at the Longji site are estimated to be within the rectangle area (in light orange) framed by three orange boundary dots. Black scatter points represent bearing parameters of terrestrial soil samples, and dashed-dotted lines represent typical bearing parameters of terrains on the Moon ($N = 1.0$, $K_s = 820$ kPa m$^{-N}$). LLL represents Land Locomotion Lab. WES represents Waterways Experiments Stn. $N$ is the variable sinkage exponent of wheel-terrain interaction; for black scatter points and dashed-dotted lines, $N = n$ (intrinsic sinkage exponent of the terrain). The bearing and shearing parameters of terrestrial soil samples are listed in Supplementary Table 9. **b** Lunar regolith shearing parameters analysis in comparison to terrestrial soil samples. The shearing parameters of lunar regolith at the Longji site are estimated to be within the rectangle area (in light orange) framed by two orange boundary dots. Black scatter points represent shearing parameters of terrestrial soil samples, and dashed-dotted lines represent typical shearing parameters of terrains on the Moon ($c = 520$ Pa, $\varphi = 42°$). **c** Lunar regolith shearing

parameters of the Longji site compared with those estimated in other lunar exploration missions. The shearing parameters of soil samples on other lunar landing sites are listed in Supplementary Table 10. Each error bar defines the upper and lower bound of $c$ and $\varphi$ estimated in different lunar exploration missions with different methods, and the center point is the median value of the parameter range. **d** Lunar regolith lateral parameters analysis in comparison to the lunar simulants. The shearing parameters of lunar regolith at the Longji site are estimated to be within the rectangle area (in light orange) framed by two orange boundary dots. Dashed-dotted lines represent the lower and upper bound of $K_y$ according to experience ($K_y = 13$ mm, $K_y = 40$ mm). The lunar simulant is equivalent to FJS-1[85], whose material components and mechanical characteristics (Supplementary Table 12) are similar to those of the real lunar soil, as reported in ref. 49. Another simulant (Supplementary Table 13) is standard commercial experimental sand, numbered HIT-LSS2, whose particle size is set to be uniform for repeatability of terramechanics experiments. The lateral parameters of the lunar simulant and the standard commercial experimental sand are from refs. 43,86.

We also tried to estimate the lateral parameters (specifically refer to lateral shear parameters) for the farside regolith in the extra-terrestrial environment. The lateral properties of the lunar regolith characterize the lateral force on the wheel during the wheel-terrain interaction when the wheel has a lateral traveling velocity $v_y$. The properties are parameterized by the external friction angle $\varphi_y$ and the lateral shearing deformation modulus of the soil $K_y$. The external friction angle $\varphi_y$ represents the roughness between the wheel surface and the regolith, and the lateral shearing deformation modulus of the soil $K_y$ represents the tangential shear strength of the regolith. Here, we provide a preliminary result of the lateral property of the lunar regolith investigated via in-situ data. The lateral force (denoted by $F_L$) on the lateral terramechanics models was deduced from the dynamic

simulation, so that the lateral characteristic curves under different side slip angles (ranging from 0.25° to 10°) can be predicted as shown in Fig. 4d. Taking 13 mm (the typical value of lunar soil simulant) and 40 mm as the lower and upper bound of the lateral shearing deformation modulus (denoted by $K_y$) respectively, for the measured side slip angle (denoted by $\beta$) within 0.5°–3° during the traverse, the estimated external friction angle (denoted by $\varphi_y$) was restricted to 8.3°–16.5° enclosed by curves. The longitudinal and lateral shear properties represent the shear strength of the soil and limits the maximum drawbar pull to the wheel when traverse on regolith.

Benefiting from the digital twin system, unmeasurable data in the missions can be reproduced, making it possible to use more accurate models requiring much data obtained for identification and identify

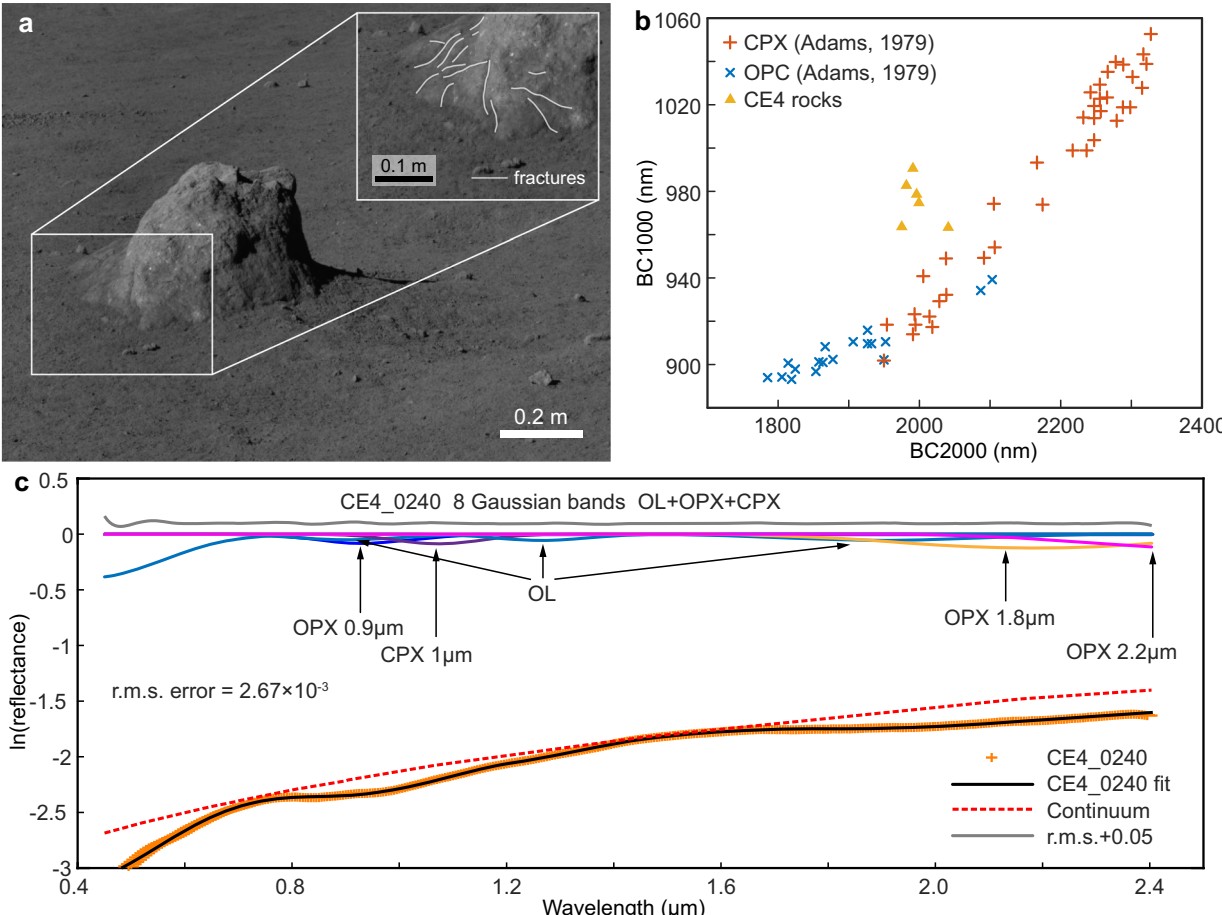

**Fig. 5 | Surface morphology and reflectance spectra of the rock investigated.**
**a** Part of the visible light image of the target rock taken with the Pancam on the 41st lunar day. The spectra associated with the image range from 420 to 700 nm. The inset shows several fractures on in the lower part of the rock. **b** Comparison of the 1 and 2 µm band center positions of the CE-4 rocks with pyroxenes measured by Adams (1974)[52]. **c** Rock D41N240 spectra modeled with the OL-OPX-CPX configuration by the MGM. Source data are provided as a Source Data file.

parameters closer to the ground truth. Exemplified by the regolith property identification, the digital twin originated in engineering[51] has gone beyond and emerges as a potent asset in planetary science[52]. The parameter identification results were further used to predict the backhaul movement of Yutu-2 (Supplementary Fig. 8), which indicated a higher prediction accuracy of about 0.09 m offset at the endpoint relative to the primary unrectified counterpart (approximately 0.44 m in offset conditioned on $\varphi = 42°$, $K_y = 20.0$ mm). The elaborate wheel-regolith interaction models, adaptable to various motion conditions, constitute the cornerstone for predicting the rover's realistic mobility behavior within the digital twin system, while the real-world data is used to update the models towards higher fidelity. The high integration of models and data within the digital system exemplified by behavior prediction demonstrate the system's remarkable capability to bridge the gap between simulation and reality, with broad implications for other sim2real challenges. This approach results in a more realistic planetary system state simulation, thereby enhancing the reliability of prediction[52].

**In-situ spectral investigations of the target rock**
Close-focused investigation shows that the target boulder (Fig. 5a) with long strips is embedded in fine-grained lunar soil, with a height of approximately 0.41 m. It has an irregular surface morphology, partially arc-shaped and with some sharp rock edges. On the basis of the Pancam images taken by the Yutu-2 rover, several fractures (with widths and lengths of 0.05–0.10 m and 0.2 m, respectively) are evident in the

lower part of the boulder. The boulder has a relatively large size and is surrounded by various irregular fragments, which implies that the effects of space weathering on these fragments may be limited compared with those for lunar regolith. In addition, several cracks on the boulder surface indicate that it might have suffered certain degrees of impact.

Fundamental spectral parameters were calculated to estimate the boulder composition after the spectrum preprocessing procedure (described in the Methods subsection spectrum band parameter and modified Gauss model (MGM)). Six rock targets investigated previously with addition of this target boulder are shown in Supplementary Fig. 9a. Band center positions of 1000 nm (abbreviated as BC1000) and 2000 nm (abbreviated as BC2000) absorption features were derived to identify low-Ca pyroxenes (about 900 nm and 1900 nm) and high-Ca pyroxenes (about 1100 nm and 2200 nm) (Fig. 5b). In principle, with increasing iron and calcium content, orthopyroxene's absorption band center shifts to longer wavelengths, while with increasing magnesium content, it shifts to shorter wavelengths. Meanwhile, the absorption band center of clinopyroxene shifts to longer wavelengths with increasing calcium, and to shorter wavelengths with increasing iron[53–56]. Here, the parameters of the six rock targets were assembled, and the results indicated that they were homologous and might share a similar region or source (Fig. 5b). The target boulder was inferred to be composed of Fe/Mg-rich low-Ca pyroxene with a shorter BC2000 and shorter BC1000, consistent with the Mg-pyroxene annulus region[19].

In addition to the fundamental spectral parameters, the modified Gauss model, which was sensitive to mafic mineral assemblages (https://sites.brown.edu/relab/mgm/), was also used to estimate the relative abundance of olive (OL), orthopyroxene (OPX) and clinopyroxene (CPX). MGM fitting with OL-OPX-CPX provided the optimal modeling for the rock spectra. Figure 4c shows the modeling result of the target boulder (i.e., D41N240), and the modeling results of all six rocks at the CE-4 sites are shown in Supplementary Fig. 9c, d. Based on the MGM results and band strength, there is a correlation between the normalized band strength ratio (NBSR) and pyroxene-olivine mineral fraction[57,58]. The NBSR results of all six rocks are listed in Supplementary Table 2. The average relative abundance between OPX and CPX is 46% versus 54%, and the relative abundance of olivine is in the range of 20-30%. The rocks D13N106 and D23N150 are slightly offset from the range potentially due to heavier space weathering and shadow effects, recognized from the Supplementary Fig. 9a. We estimated the fractions of olivine and pyroxene (OPX and CPX) and accordingly their origins based on the MGM and NBSR results. The abundances of minerals (i.e., olivine, OPX, and CPX), as indicated by the mosaic map created from topographically corrected Mineral Mapper reflectance data from the Kaguya Multiband Imager (MI), is similar to the modal mineralogy of ejecta from Zhinyu crater (Supplementary Table 3). Note that the fitting error for the MGM method was 0.267% and since plagioclase was not included in the MGM methods, only relative abundance of mafic olivine, OPX, and CPX were compared with the orbiter-derived data. We understand that orbital-derived and in-situ-derived spectral data are difficult to compare due to the difference in spatial scales directly; however, because the in-situ spectra suggest the fin-shaped rock is consistent with previously investigated rock targets along the traverse and in agreement with the derived mineralogical compositions of the Zhinyu crater, we infer that the target boulder belongs to the Zhinyu crater ejecta rather than those of the Finsen crater. Further investigations along the traverse of the Yutu-2 rover and potentially other sample-returned missions would be required to constrain the precise source regions of the local rocks.

## Discussion

Digital twin-based analysis carried out in this study enabled the expedition of Yutu-2 rover in high skidding risk along a narrow and uneven passage, and successfully got additional geological insights of the lunar farside characteristics regarding the regolith property and the source of the surface rock. Precise locomotion simulation and thorough pre-traverse evaluation significantly mitigate the risks of mission failure by meticulously modeling wheel-terrain interactions and leveraging real-world data. The experience in locomotion and interaction analyzed in this study yields valuable data for regolith mechanics studies, offering insights into how lunar regolith responds to a variety of static and dynamic loadings. This knowledge is essential for the development of terramechanics models that underpin interactions with rovers, landers, scoops, and other vehicles on the lunar surface[21].

In this study, the lunar farside regolith at Chang'E-4 site was characterized to be stronger than the counterpart of Apollo missions and close to the terrestrial sandy loam in Michigan in bearing strength. We estimate its cohesion parameter of the shear property to be about 520-3154 Pa with associated internal friction angle of 21.5°–42.0°, indicating larger cohesion to most nearside lunar regolith measured. Besides, the lateral property of the lunar regolith was also investigated here using terramechanics models under lateral slip states and the estimated external friction angle was within 8.3°–16.5°. The in-situ identification based on wheel-terrain interaction models with the support of digital twin filled the gaps of the shear property estimation of the lunar farside regolith, and also provided a promising method to infer lateral properties of the regolith in extraterrestrial environments with initial parameters. The tri-aspect

parameter identification results presented here serves as a foundational step for subsequent engineering verifications in the lunar farside, such as the assessments of landing feasibility, rover mobility, and sample excavation capabilities. Moreover, these findings may have implications for future in-situ resource utilization (ISRU) technologies[59], where lunar regolith plays a pivotal role in base construction, mining, and resource extraction.

In our analysis, the fin-shaped rock was homologous in spectral parameters to six rocks investigated previously in the Chang'E-4 mission and inferred to be sourced from the Zhiyu crater ejecta based on its relative abundance of OL-OPX-CPX assemblages. This is the second large rock fragment inferred to be ejected from the Zhinyu crater along the traverse of the Yutu-2 rover. Zhinyu crater is a relatively young impact crater located about 30 km west of the landing site[60]. We found that the rover had closely investigated a previous rock fragment on the third lunar day and concluded it originated from the Zhinyu crater[25]. Therefore, our discovery of the lastest rock piece originated from Zhinyu cater supports that besides the majority of ejecta blanket from the Finsen crater to the CE-4 landing site, other source regions may also contribute to the local materials, suggesting a more complicated geological history in the local area. Meanwhile, the fin-shaped rock is also dominant by plagioclase, consistent with the previous suggestions that the CE-4 region mainly sampled the mare basalt instead of deeper mantle materials[60]. These in-situ observations provide important ground truth for remote sensing investigation of the region and lay an essential foundation for reconstructing the thermal evolution history of the regional mare activities, and provide a better understanding of the geological context of SPA and the compositions of the lunar dichotomy at the farside, as well as the upcoming CE-6 sample-return mission[61].

Even though the identification methods and results on regolith mechanical properties at the Longji site provide a representative sample for investigating and understanding the regolith properties in the lunar farside, it is a localized result. More samples at other sites (e.g., highland, polar regions) need to be investigated to supplement the full picture of lunar regolith among different terrains and regions. In addition to the macroscopic properties of the lunar soil, its microscopic properties are also worth an in-depth examination, which can be achieved through discrete element-based wheel-regolith interaction simulations[62]. The methodology we employ for lunar regolith parameter identification is rooted in wheel-terrain interactions, consequently, it is ineffective on hazardous terrains, where the planetary rover cannot traverse successfully, such as soft sand dunes or inner crater walls. To get rid of the reliance on mobility experience, it is worth exploring alternative regolith interaction tools, such as bucket[63] and robotic arms[64], and their interaction models for parameter identification. These tools can be utilized for preliminary haptic inspections, allowing for obtaining interaction data without exposing the rover to potential mobility risks. By integrating the haptic results with visual information, we can expand the scope of prediction to a larger front area and use it for planning[65].

In addition, the whole locomotion and rock investigation process demonstrated a hidden development logic and an inseparable relationship between the extraterrestrial geological science and robotic exploration technology, where planetary science was enabled by engineering support, and results of scientific exploration in turn fed back to the improvement of technology towards higher fidelity and reliability, exemplified by the evolution of the digital twin system during the mission. To extend the bounder of extraterrestrial geological research, breakthrough on the modern robotics technology is expected to happen in multiple aspects, including exploration robots with more adaptive locomotion capability, more variable payloads, and more advanced intelligent system. With the powerful technique support, it will be possible to explore previously

inaccessible sites, such as caves, lava tubes, and permanent shadow regions, which are scientifically valuable but challenging for robots. Exploration on such sites have the potential to uncover unknown habitations, available resource and deepen humanity's understanding of the space environment as well as the evolution of the Solar system.

## Methods

### Instruments and data description

The Yutu-2 rover (Supplementary Fig. 2b) is a six-wheeled robot designed in a rocker-bogie suspension, which is similar to National Aeronautics and Space Administration (NASA)'s series of Mars rovers (Spirit, Opportunity[66], Curiosity (https://mars.nasa.gov/msl/spacecraft/rover/wheels/), and Perseverance (https://mars.nasa.gov/mars2020/spacecraft/rover/wheels/)) in suspension configuration. An internal differential mechanism connects the left and right rocker bogies assemblies to the rest of the vehicle, which is different from rovers in Mars Science Laboratory (MSL)[67] and Mars 2020[68] missions, whose differential mechanism are outside of the body connecting to the left and right rockers and to the rover body by a pivot in the center of the rover's top desk. Compared with other mobility architectures, the rocker-bogie suspension enjoys several advantages[66]: (1) equilibrated ground pressure on wheels for consistent weight distribution; (2) continuous contact of all six wheels on rugged terrain, aiding propulsion.; (3) stabilized body angle with differential mechanism, providing a steady instrument platform; (4) absorbing driving load energy for enhanced structural durability. All six driving wheels are independently driven, and only the front two wheels and rear two wheels have additional steering degree of freedom. There is a steering offset that is the distance between the steering axis and the front and rear wheels. Each wheel on Yutu-2 rover is equipped with an aluminum alloy hub attached with mesh and 24 grousers evenly arranged in two staggered rows, which is different from wheels on NASA's Mars rovers in design, as illustrated in Supplementary Fig. 10. In such a specific configuration, Yutu-2's wheels are light-weighted while having sufficient loading capability and great traction performance. Yutu-2 rover can climb up slopes no more than 20°and it is also able to surmount obstacles up to 200 mm in height. The rover has seven working modes to cope with different environments and adapt to various locomotion conditions. The parameters of the rover and its grouser wheels are listed in Supplementary Table 4.

The data used in this study include images from a Pancam, Navcam, and two HazCams, alongside the visible and near-infrared (VNIR) hyperspectral images and single-pixel short-wave infrared (SWIR) spectra from the visible and near-infrared imaging spectrometer (VNIS) instrument[7].

Pancam is one of the scientific payloads used primarily for high-resolution mapping and localization. It is mounted on the mast of the Yutu-2 rover, consisting of two optical systems of identical functions, performances, and interfaces, parameterized as Supplementary Table 5. In this study, the Pancam stereo images are the data source for generating high-resolution topographic maps of the Longji site surface and determining Yutu-2's position using photogrammetric techniques. Its imagery is also used to infer the physical properties of the lunar regolith, construct the shape model of the rock target, and measure its geometric parameters. Navcam and HazCam are two engineering payloads onboard Yutu-2, providing color and grayscale images used for navigation and hazard detection, respectively. Both the front and back sides of the rover's carriage are equipped with a pair of Hazcams. The parameters of the Navcam and Hazcam can also be found in Supplementary Table 5.

The VNIS[69,70] is composed of a complementary metal-oxide-semiconductor (CMOS) image with 256 by 256 pixels, a SWIR spectrometer, and a white panel for calibration and dust-proofing. It is assembled on the anterior of the rover to detect the composition of the lunar surface materials at a fixed 45° zenith angle at a height of 0.69 m. Its spectral wavelength ranges are 450–945 nm and 900–2395 nm with a default sampling interval of 5 nm. For the SWIR spectrometer, its field of view is a circular area in the CMOS image centered at (96, 128) with a 54-pixel radius (Supplementary Table 6). The measurement uncertainties are 5% and 7% for visible (VIS)/near-infrared (NIR) and SWIR, respectively[69].

### Wheel-terrain interaction modeling

For the Yutu-2 rover, each grouser wheel moving with an angular velocity $\omega$ on the lunar regolith can be individually modeled based on terramechanics principles. When the wheel moves with steering or suffers side slippage, its forward velocity $v$ deviates from its heading direction $v_x$ due to the existence of lateral traveling velocity $v_y$ at a degree, which is defined as the side slip angle $\beta$, as shown in Supplementary Fig. 3a. Each wheel under its own gravitational pressure is applied with external forces and torques (the wheel load $W$, the resistance force $f_{DP}$ applied from the rover suspension, and the driving torque $T$ applied at the wheel rotation axis by the actuator) from the rover body, as well as the continuously distributed stress (normal stress $\sigma$ and shear stress $\tau$) from the lunar regolith, as shown in Supplementary Fig. 3b-d. Due to the existence of the side slip angle $\beta$, the shear stress of wheel is composed of the longitudinal shear stress $\tau_x$ and the lateral shear stress $\tau_y$. As a result, the wheel is also subjected to an additional lateral force $F_L$ acting along the lateral direction of the wheel.

The integration effect of the distributed normal stress $\sigma$ and the longitudinal shear stress $\tau_x$ along the wheel-terrain interaction region (from the entrance angle $\theta_1$ to the exit angle $\theta_2$) is presented as the normal force $F_N$, the drawbar pull $F_{DP}$ and the driving resistance torque $M_R$ on the wheel in the macroscopic view. Therefore, these forces are modeled as follows:

$$F_N = rb \int_{\theta_2}^{\theta_1} \sigma(\theta) \cos\theta \, d\theta + r_s b \int_{\theta_2}^{\theta_m} \tau_x(\theta) \sin\theta \, d\theta = W \qquad (1)$$

$$F_{DP} = -rb \int_{\theta_2}^{\theta_1} \sigma(\theta) \sin\theta \, d\theta + r_s b \int_{\theta_2}^{\theta_1} \tau_x(\theta) \cos\theta \, d\theta = f_{DP}, \qquad (2)$$

$$M_R = r_s^2 b \int_{\theta_2}^{\theta_1} \tau_x(\theta) \, d\theta = T, \qquad (3)$$

where $r$ is the wheel radius, $r_s$ is the equivalent shearing radius of the grouser wheel and $b$ is the width of the wheel.

The equivalent shearing radius $r_s$[33] can be calculated as

$$r_s = r + \lambda_s h (0 \le \lambda_s \le 1), \qquad (4)$$

where $\lambda_s$ is the lug shearing coefficient valued within 0.0–1.0, and its typical value is 0.5.

The wheel's lateral force $F_L$ is usually considered to consist of the force $F_u$ beneath the wheel due to shear motion, and the resistance force $F_s$ acting on the side face of the wheel[43]. The force $F_u$ is produced by $\tau_y(\theta)$ beneath the wheel, while $F_s$ is the reaction force generated by the bulldozing phenomenon on the side face of the wheel[43]. The resistance force $F_s$ is relatively small, which is almost 5%–10% of the lateral force $F_L$[71], and usually is neglected in the calculation. Therefore, the lateral force $F_L$ is mainly the effect of $F_u$ and can be calculated as

$$F_L = r_s b \int_{\theta_2}^{\theta_1} \tau_y(\theta) \, d\theta = F_u. \qquad (5)$$

On the basis of Ding's model[42], the normal stress $\sigma(\theta)$ in Eqs. (1–3) is expressed in detail as

$$\sigma(\theta) = \begin{cases} \left(\frac{k_c}{b} + k_\varphi\right) r^N (\cos\theta - \cos\theta_1)^N (\theta_m \le \theta \le \theta_1) \\ \left(\frac{k_c}{b} + k_\varphi\right) r^N \left\{ \cos\left[\theta_1 - \frac{\theta - \theta_2}{\theta_m - \theta_2}(\theta_1 - \theta_m)\right] - \cos\theta_1 \right\}^N (\theta_2 \le \theta < \theta_m) \end{cases},$$

$$(6)$$

where $k_c$ is the cohesive modulus of the soil, $k_\varphi$ is the frictional modulus of the soil, $N$ is the variable sinkage exponent of the wheel-terrain interaction. The term $k_c/b + k_\varphi$ can be denoted as $K_s$, the equivalent stiffness modulus of the soil. The variable sinkage exponent $N$ correlated with the slip ratio $s$ is represented by a linear function as[42]

$$N = n_0 + n_1 s, \qquad (7)$$

where $n_0$ represents the static sinkage exponent and $n_1$ represents the dynamic sinkage exponent reflecting the effect of the slip ratio to the wheel sinkage. Previously, the dynamic sinkage caused by slip ratio was not considered; thus, it was deemed that $N = n$, where $n$ was the intrinsic sinkage exponent of the terrain. The slip ratio of a wheel $s$ is defined as follows:

$$s = \begin{cases} 1 - v/(r_s\omega)(r_s\omega \ge v, 0 \le s \le 1) \\ (r_s\omega)/v - 1(r_s\omega < v, -1 \le s < 0) \end{cases}. \qquad (8)$$

If $s > 0$, the wheel is in slip condition; if $s = 0$, the wheel rolls without slipping and skidding; and if $s < 0$, the wheel is in skid condition.

The entrance angle $\theta_1$, the exit angle $\theta_2$, and the maximum stress angle $\theta_m$ can be computed as

$$\theta_1 = \arccos[(r - z)/r], \qquad (9)$$

$$\theta_2 = c_3\theta_1, \qquad (10)$$

$$\theta_m = (c_1 + c_2 s)\theta_1, \qquad (11)$$

where $z$ is the wheel sinkage, $c_1$, and $c_2$ are the coefficients of the wheel-terrain interaction angle, and $c_3$ is the coefficient of the exit angle, characterizing the ratio of the exit angle $\theta_2$ to the entrance angle $\theta_1$. In the calculation, $c_1$ and $c_2$ are set to 0.5 and 0, respectively, and the value of $\theta_m$ is equal to 0.5 times of $\theta_1$. The coefficient $c_3$ is taken as zero in the calculation.

Based on the Janosi equation[72], we improved the conventional longitudinal shear stress $\tau_x$ to accommodate both the wheel slip and skid states as

$$\tau_x(\theta) = \mathrm{sgn}(s) \cdot [c + \sigma(\theta)\tan\varphi](1 - e^{-j/K_x}), \qquad (12)$$

$$\mathrm{sgn}(s) = \begin{cases} -1(-1 < s < 0) \\ 1(0 \le s < 1) \end{cases}, \qquad (13)$$

where $c$, $\varphi$, $j$, and $K_x$ are the cohesion, the internal friction angle, the shearing deformation, and the longitudinal shearing deformation modulus of the soil, respectively. The main difference between the wheel slip and skid states for the calculation of $\tau_x$ is shown in the distinct expression of the longitudinal shearing deformation $j$[73], which is expressed as

$$j(\theta) = \begin{cases} r_s[(\theta'_1 - \theta) - (1 - s)(\sin\theta'_1 - \sin\theta)](0 \le s < 1) \\ r_s[(\theta'_1 - \theta) - \frac{1}{1+s}(\sin\theta'_1 - \sin\theta)](-1 \le s < 0) \end{cases}, \qquad (14)$$

where $\theta'_1$ is the equivalent entrance angle. The equivalent entrance angle $\theta'_1$ of a grouser wheel[42] can be calculated as

$$\theta'_1 = \arccos[(r - z)/(r + h)] \qquad (15)$$

where $h$ is the grouser height.

The lateral shear stress $\tau_y$ is modeled in the same fashion as $\tau_x$ as

$$\tau_y(\theta) = [c + \sigma(\theta)\tan\varphi_y](1 - e^{-j_y/K_y}), \qquad (16)$$

where $\varphi_y$, $j_y$, and $K_y$ are the external friction angle, the lateral shearing deformation, and the lateral shearing deformation modulus of the soil, respectively. The lateral shearing deformation of the soil $j_y$ is expressed as

$$j_y(\theta) = \begin{cases} r(1 - s)(\theta'_1 - \theta)\tan\beta(0 \le s < 1) \\ [r/(1+s)](\theta'_1 - \theta)\tan\beta(-1 \le s < 0) \end{cases}, \qquad (17)$$

where $\beta$ is the side slip angle.

When $\theta$ comes to $\theta_m$, the corresponding normal and shear stress reach their maximum as

$$\sigma_m = \left(\frac{k_c}{b} + k_\varphi\right) r^N (\cos\theta_m - \cos\theta_1)^N, \qquad (18)$$

$$\tau_{xm} = (c + \sigma_m \tan\varphi) \times \left(1 - e^{-j/K_x}\right) \qquad (19)$$

$$\tau_{ym} = (c + \sigma_m \tan\varphi_y) \times (1 - e^{-j_y/K_y}). \qquad (20)$$

Although the integration expressions of $F_N$, $F_{DP}$, $M_R$, and $F_L$ are accurate, they are not cost-effective in calculation. Therefore, an appropriate linearization method[74] is used to simplify Eqs. (1a–c, 3) into a more compact and unified form for implementation as follows:

$$F_N = rb\sigma_m A + \mathrm{sgn}(s) \cdot r_s b\tau_{xm} B, \qquad (21)$$

$$F_{DP} = \mathrm{sgn}(s) \cdot r_s b\tau_{xm} A - rb\sigma_m B, \qquad (22)$$

$$M_R = \mathrm{sgn}(s) \cdot r_s^2 b\tau_{xm} C, \qquad (23)$$

$$F_L = r_s b\tau_{ym} C, \qquad (24)$$

where

$$A = \frac{\cos\theta_m - \cos\theta_2}{\theta_m - \theta_2} + \frac{\cos\theta_m - \cos\theta_1}{\theta_1 - \theta_m},$$

$$B = \frac{\sin\theta_m - \sin\theta_2}{\theta_m - \theta_2} + \frac{\sin\theta_m - \sin\theta_1}{\theta_1 - \theta_m},$$

$$C = (\theta_1 - \theta_2)/2.$$

In our models, the terrain mechanical parameters are characterized in bearing, longitudinal shearing and lateral shearing three aspects. The bearing properties are related to the bearing strength of the soil represented in $K_s$ and $N$; while the shearing properties regarding soil shear strength are characterized in the longitudinal and tangential directions. They are symbolized by $c$, $\varphi$, $K_x$, and $\varphi_y$, $K_y$, respectively. The external friction angle $\varphi_y$ represents the roughness between the wheel surface and the soil, and the lateral shearing

deformation modulus $K_y$ represents the tangential shear strength of the soil.

These terramechanics models consider the slip sinkage phenomena and the wheel-lug effect, and could predict the normal force, resistance moment, and drawbar pull with higher accuracy compared to conventional terramechanics models[75], and their modeling error is less than 10%[42] as illustrated in previous studies. The bearing model (Eq. (1)) is adopted from Bekker model[76] with semi-empirical equations, thus it is based on the assumption that the wheel slip ratio is no larger than 0.6[71]. For high slippery ($s > 0.6$) cases, the accuracy of the above semi-empirical terramechanics models will get significantly affected, and cannot reach a reasonable agreement with the wheel mobility performance. In this study, the rover wheel work on skid condition during the outbound traverse, and most the wheel slip ratios are between 0 and −0.075 and no less than −0.1, satisfying the slip ratio assumption and can be applied to parameter identification. When the slip ratio is over 0.6, the parameter identification process can still obtain a set of parameters in the sense of fitting, but the model is no longer in high accuracy in such condition, thus the accuracy of the identified parameter is also affected.

### Lunar regolith parameter estimation at the Longji site

Lunar regolith parameters are estimated based on terramechanics models to be compatible with the collected locomotion data as shown in Supplementary Fig. 11. We note that Eqs. (21) and (22) are expressions of $F_N$ and $F_{DP}$, and they are both about parameters of $\{\sigma_m, \tau_{xm}, s, z, r, b, r_s\}$. In these two equations, the only unknown variables are $\sigma_m$ and $\tau_{xm}$, while other variables can be measured or obtained according to the system configuration. On the other hand, Eq. (24) shows that $F_L = f_L(\tau_{ym}, z, r, b, r_s)$, in which the only variable remaining unknown is $\tau_{ym}$. Thus, stress variables, $\sigma_m$, $\tau_{xm}$, and $\tau_{ym}$, representing the normal stress, longitudinal shear stress, and lateral shear stress during the wheel-terrain interaction, can be solved in these three equations, acting as a bridge to estimate mechanical parameters of the farside lunar regolith. Regarding the normal bearing parameters of the lunar regolith, we found that the solved $\sigma_m$ is an expression represented by $\{z, K_s, N, r\}$ according to Eqs. (18), (7) and (9). Then, the relationship between the sinkage exponent $N$ and the equivalent stiffness modulus $K_s$ under different wheel sinkage $z$ can be plotted as shown in Fig. 4a because that the wheel radius $r$ is known. For the measured wheel average sinkage $z$ of 8 mm, the estimated bearing property of the lunar regolith is most likely conditioned on the yellow curve ($z = 8$ mm) and between points ($N = 0.89$, $K_s = 820$ kPa m$^{-N}$; $N = 1.0$, $K_s = 1443$ kPa m$^{-N}$), which are determined by setting one of the $K_s$ and $N$ as the typical value. For the wheel sinkage of 5–15 mm, the estimated $K_s$ and $N$ are in the orange region. On the other hand, the solved $\tau_{xm}$ is an expression about $\{\sigma_m, s, z, c, \varphi, K_x, r_s\}$, in which only the soil shearing parameters $\{c, \varphi, K_x\}$ remain unknown. The $K_x$ is a non-dominant one among the shearing parameters and its variation has limited effect on the characteristics of the model, thus we set it as a fixed value of 17.8 mm (its typical value). Then, the shearing characteristic curves of $c$ and $\varphi$ under different slip ratio $s$ can be plotted as shown in Fig. 4b. In regard to the lateral characteristics, the relationship between the lateral shear deformation modulus $K_y$ and the external friction angle $\varphi_y$ can be built upon the expression $\tau_{ym} = f_{rm}(\sigma_m, s, z, \beta, c, \varphi_y, K_y, r)$ according to Eq. (20). In this expression, $\tau_{ym}$ can be solved in Eq. (24), and the wheel states $\{s, z, \beta\}$ are measurable, and the remaining parameters are known or determined above. Lateral characteristic curves of the lunar regolith are shown as Fig. 4d.

As explicitly shown in Supplementary Fig. 11, inputs into these models for lunar regolith parameter estimation include forces ($F_N$, $F_{DP}$, $F_L$), wheel states (slip ratio $s$, sinkage $z$, side slip angle $\beta$), and wheel parameters ($r$, $b$, $r_s$). In this study, the normal force $F_N$, the drawbar pull $F_{DP}$, and the lateral force $F_L$ are solved through the constrained dynamics of a multi-rigid-body system with a floating base in the digital

twin. The relative errors of estimated forces compared to experimental measurements are within 10%[40,43], as illustrated in previous studies. The wheel sinkage $z$ is calculated by comparing the elevation difference of the track area and the surrounding surface plane in the reconstructed digital elevation model, which is reconstructed using stereo Navcam images in 1024 × 1024 pixels. The accuracy of the wheel sinkage measurement is dependent on the accuracy of the single-site digital elevation model (DEM), whose topographic data can reach the submillimeter level[50] (i.e., its error is within 1 mm). The wheel slip ratio $s$ is solved with the actual travel distance of Yutu-2 estimated by visual localization and with the planned travel distance by rearranging Eq. (8). The accuracy of the slip ratio is dependent on the visual localization yielding an estimation error of 4%[39]. The wheel side slip angle $\beta$ is estimated using the posture of the rover, and the side slip angle along the forward traverse varied between 0.5° and 3°, whose accuracy depends on the visual positioning and its estimation error is 4%[41]. The remaining rover wheel parameters $r$ and $h$ are constant depending on the rover configuration as listed in Supplementary Table 4, and the equivalent shearing radius $r_s$ for Yutu-2's grouser wheel is calculated according to Eq. (4) with lug shearing coefficient of 0.65. The values or ranges of the inputs used in this study are in Supplementary Table 7.

In this context, the primary source of error in identification outcomes stems from the measurement uncertainties associated with input variables, including slip ratio, sinkage, side slip angle, and interaction forces. Due to the complexity and nonlinearity of our terramechanics models, getting an analytical uncertainty analysis of model output is a difficult task. Therefore, we use Monte Carlo simulations[77] to calculate the uncertainty of model outputs by considering input measurement uncertainties. These measurement errors are represented as normal distributions. The Monte Carlo simulation involves a sample size of 1024, which has been assessed as sufficient for obtaining reliable and robust results[78]. Regarding bearing properties, when we assume a sinkage exponent ($N$) of 1.0, as the typical value from Apollo missions, the equivalent stiffness ($K_s$) ranged from 556.7 to 1443 kPa m$^{-N}$ for estimated sinkage of 8–15 mm. In other traverses, wheel sinkage may be smaller but generally not less than 5 mm. For very shallow sinkage, $K_s$ approaches its upper limit with minimal adjustability. Consequently, for a sinkage of 5 mm, $K_s$ was fixed at 1443 kPa m$^{-N}$. To account for the measurement error associated with wheel sinkage, which was formulated as a normal distribution with a mean of 5 mm and a standard deviation of 0.33 mm (following the three-sigma rule), we conducted Monte Carlo simulations to estimate the corresponding distribution of output sinkage exponent $N$. The mean of the estimated $N$ value is 0.87 with a standard deviation of 0.017 when only considering sinkage measurement uncertainties. When considering uncertainties in forces ($F_N$, $F_{DP}$) without accounting for the uncertainty of $z$ (wheel sinkage), the standard deviation of the estimated $N$ is 0.006, indicating that the predicted $N$ is more sensitive to the wheel sinkage errors than to force errors in this case. By considering both uncertainties in forces and wheel sinkage, the standard deviation of the estimated $N$ is estimated to be 0.017, as illustrated in Supplementary Fig. 13(a). For the largest sinkage of 15 mm, the standard deviation of the estimated $K_s$ is approximately 25.8 kPa m$^{-N}$ (Supplementary Fig. 13(b)) when the sinkage exponent $N$ is at the typical value of 1.0. Concerning shear properties, with a slip ratio of −0.075 and accounting for measurement errors on forces and slip ratio, the cohesion was determined to be 520 Pa (the typical values of the lunar regolith), and the internal friction angle $\varphi$ was estimated to be 21.5° with a standard deviation of 2.29°, as illustrated in Supplementary Fig. 13(c).

Admittedly, the terrain is assumed to be homogeneous and the regolith parameters are assumed to be constant for the above parameter identification process. Since the traverse data used in this study is carried out within a small region (within 20 m²), and there is no significant difference on the surface regolith according to the

observation, the regolith mechanics should be almost consistent, and our identification results should be representative. However, such assumption is not always valid, especially on larger regions or on regolith-covered surface in different grain sizes. In case the regolith mechanics varied from place to place, it will be necessary to update terrain parameters according to the variations in the interaction states[43].

## Rover locomotion simulation

The surface of the Chang'E-4 landing site is covered with approximately 12 m thick of unconsolidated regolith[79], produced by the long-term impact gardening and space weathering of the lunar rocks. Moving on such soft and deformable lunar regolith, the rover will suffer peculiar wheel slippage and sinking, which is hard to simulate using conventional simulation systems mainly designed based on rigid body dynamics. Therefore, a locomotion simulation considering the interaction between the wheel and the soft terrain is carried out in this work to predict the dynamic behavior of the Yutu-2 rover moving on such terrain.

A general overview of the dynamic simulation system is shown in Supplementary Fig. 4. The Yutu-2 rover is modeled as a rigid floating base with multiple wheels, and each wheel is constrained with the fore-established wheel-terrain interaction forces and torques. The DEM of the terrain containing both geometric and physical parameters, the Yutu-2 dynamic model, and movement commands as well as the control model in gray boxes are prepared and provided as input. The core of the dynamic simulation mainly lies in two modules, as these two red boxes shown in Supplementary Fig. 4. One is the wheel-terrain interaction states solution, and the other is the interaction forces and torques solution. The former determines the terrain geometric and mechanical parameters of the interaction area, and key wheel states (the sinkage $z$, the slip ratio $s$, and the side slip angle $\beta$ of the wheel). The latter calculates the interactive forces and torques based on the fore-established terramechanic models. Then, with the obtained interactive forces, torques and control model, the rover's position, velocity, and acceleration can be updated according to the dynamics model of the rover, which is solved using the Vortex dynamic engine[80]. The above-mentioned processes make up a single-step iteration of the simulation, and continuous iterations present the whole simulated locomotion of the Yutu-2 rover on the lunar regolith.

**Wheel-terrain interaction states solution.** For a grouser wheel moving on deformable terrains, its wheel-terrain interaction area is shown as the left red box in Supplementary Fig. 4. In this figure, $\{\Sigma_I\}$ is the inertial frame; $\{\Sigma_e\}$ is the wheel-terrain interaction frame. $\{\Sigma_c\}$ is the wheel center frame, and its origin is on the wheel center and its orientation is the same as the interaction frame. To determine the wheel-terrain interaction plane, three points ($P_1(x_1, y_1, z_1)$, $P_2(x_2, y_2, z_2)$, and $P_3(x_3, y_3, z_3)$) both on the plane and at the edge of the wheel surface are selected for calculation. $P_2$, and $P_3$ are on the outer hoops of the wheel and coincide with the entrance angle, while $P_1$ is on the middle hoop of the wheel coinciding with the exit angle. The positions of $P_1$, $P_2$, and $P_3$ are calculated using the rotation matrix $R_e$ of the wheel-terrain interaction frame $\{\Sigma_e\}$ relative to the inertial frame $\{\Sigma_I\}$ at the last time step, and the position of the wheel center $p_c$ at this time step. Therefore, the corresponding homogeneous transform matrix $T_c$ for calculation is represented in

$$\mathbf{T}_c = \begin{bmatrix} \mathbf{R}_e & \mathbf{p}_c \\ \mathbf{0}_{1\times 3} & 1 \end{bmatrix}. \tag{25}$$

where $\mathbf{0}_{1\times 3}$ is a zero matrix with one row and three columns.

The terrain map is usually divided into triangle meshes, such as $A_1$-$A_2$-$A_3$, $A_1$-$A_3$-$A_4$. Each node of the triangular mesh has associated geometric and mechanical parameters. If the contact point $P_i$ ($i = 1, 2, 3$)

lies in such a mesh, then the normal coordinate information of point $P_i$ can be obtained by linearizing and interpolating as

$$\mathbf{P}_i = (1 - u - v)\mathbf{A}_1 + u\mathbf{A}_2 + v\mathbf{A}_3, \tag{26}$$

in which (1-$u$-$v$), $u$, and $v$ are the weight coefficients of nodes $A_1$, $A_2$, and $A_3$, respectively, ranging from 0 to 1.

With these determined points on the wheel-terrain interaction plane, the wheel sinkage $z$, and the rotation matrix of $\{\Sigma_e\}$ relative to the initial frame $\{\Sigma_I\}$ at this time step used for the next iteration can be calculated according ref. 81.

The local terrain model with geometric and physical properties used in this study is a digital elevation model with physical properties named DEMP[2] proposed from ref. 38. As shown in Supplementary Fig. 12, the DEMP[2] is separated into discrete nodes and triangle elements, and each node has its own physical properties. The coordinates of terrain nodes are given as ($x, y, z, K_s, N, c, \varphi, K_x, \varphi_y, K_y$), in which ($x, y, z$) represent 3-dimensional position of the node in inertial frame, i.e., geometric properties, and ($K_s, N, c, \varphi, K_x, \varphi_y, K_y$) are physical properties of the node. The local terrain model of the Longji site was reconstructed with 0.02 m spatial resolution based on Pancam stereo images[41], and its raw geometric properties represented in elevation map is shown in Supplementary Fig. 1a. The topographic data from the single-site DEM reaches the submillimeter level[50]. In the simulation, the physical properties to be identified are under-determined, and set in a wide range of parameters to cover most cases, while non-dominant physical parameters are set as the typical value of the lunar regolith from Apollo data[49], as listed in Supplementary Table 8.

**Wheel-terrain interaction forces and torques solution.** According to the terramechanics analysis, the integral effect of the distributed stress along the wheel radius and circumference on the interaction area can be equivalent to concentrated forces and torques acting on the wheel center. As shown in the right red box in Supplementary Fig. 4, the external forces and torques acting on the wheel center include the normal force $F_N$, the drawbar pull $F_{DP}$, the lateral force $F_L$, the resistance torque $M_R$, the steering resistance torque $M_S$, and the over-turning torque $M_O$. The first four can be calculated according to Eq. (17a-d), while the last two can be calculated according to ref. 43. Therefore, the external forces and torques acting on the wheel center ($^e\!F_e$ and $^e\!M_e$) are represented as:

$$\begin{cases} ^e\boldsymbol{F}_e = {}^w\boldsymbol{F}_e = [F_{DP} \quad F_L \quad F_N]^T \\ ^e\boldsymbol{M}_e = [M_O - rF_L \quad -M_R + rF_{DP} \quad M_S]^T \end{cases}, \tag{27}$$

Then, these equivalent forces and torques in the initial frame $\{\Sigma_I\}$ ($F_e$ and $M_e$) can be calculated as:

$$\begin{cases} \boldsymbol{F}_e = \boldsymbol{R}_e{}^e\boldsymbol{F}_e \\ \boldsymbol{M}_e = \boldsymbol{R}_e{}^e\boldsymbol{M}_e \end{cases}. \tag{28}$$

## Spectrum band parameter and modified Gauss model (MGM)

The level 2B VNIS radiance data available in the Ground Research and Application System of China's Lunar and Planetary Exploration Program were further processed to obtain a reflectance factor (REFF) for spectrum analysis as follows:

$$REFF(\lambda, \theta_i) = \frac{\pi \cdot I(\lambda, \theta_i)}{E_0(\lambda) \cdot \cos(\theta_i)}, \tag{29}$$

where $\lambda$ and $\theta_i$ are the wavelength and solar incidence angle, respectively. $E_0(\lambda)$ is the solar irradiance at 1 AU. The CE-4 VNIS measurements were performed at a large phase angle (about 40°–117°) with different illumination, and photometric correction is necessary to take them to standard geometry ($i = 30°$, $e = 0°$, and $g = 30°$) using the

Lommel-Seeliger model[82]. Then, some significant dark/shadow pixels need to be removed before calculating the average VNIR reflectance in the field of view of SWIR detectors. Here, we regard the pixels with REFFs at 465 nm less than 0.02 as shadows[83]. The average VNIR reflectance is adjusted to SWIR with the overlapping points using linear regression fitting to acquire a continuous spectrum (Supplementary Fig. 9b–d). The continuous spectrum is further smoothed with 41 points using the Savitzky–Golay (SG) filter. The smoothed spectrum was removed to significant the absorption feature. The continuum line is a fitting conic with three points at 700, 1580, and 1345 nm (Supplementary Fig. 9b–d). The band center (BC) parameters are calculated from the continuously removed spectrum with the rock spectrum modeling method[84].

## Data availability

The datasets generated and analyzed during the current study are available from the corresponding author upon request. Source data are provided with this paper.

## Code availability

The code, software, procedures used in the analysis are available from the corresponding author upon request.

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

## Acknowledgements

We thank all the scientists and engineers who contributed to the Chang'E-4 mission, in particular those from the China Academy of Space Technology (CAST), Shanghai Academy of Spaceflight Technology (SAST), Harbin Institute of Technology (HIT), Beijing Aerospace Control Center (BACC) and Chinese Academy of Sciences (CAS) for their dedicated work on the Yutu-2 lunar rover and Chang'E-4 mission. This work was supported in part by: the National Key R&D Program of China under Grant 2022YFB4702300 (L.D.); the National Natural Science Foundation of China under Grant 91948202 (H.G.), Grant 51822502 (L.D.), Grant 52205011 (H.Y.); the Fundamental Research Funds for the Central Universities under Grant FRFCU9803500621 (L.D.), Grant HIT.O-CEF.2023042 (L.D.); the Natural Science Foundation of Heilongjiang under Grant LH2022E054 (H.Y.); the Heilongjiang Postdoctoral Fund under Grant LBH-Z20136 (H.Y.); and the B-type Strategic Priority Program of the Chinese Academy of Sciences under Grant XDB41000000 (Jianzhong Liu).

## Author contributions

L.D. led the overall research program; R.Zhou. conceived the main idea and prepared the manuscript with L.D. T.Y. and J.Li led the Yutu-2 rover operation, data acquisition, and analysis. H.G. led the data analysis. J.Liu led the rock scientific data analysis. Data and image analysis, locomotion simulation, and soil parameter identification: L.D., R.Zhou, H.Y., H.G., Y.Y., Z.W., H.Q., W.F., G.L., Z.D. Rover operation, data acquisition, and analysis: T.Y., X.H., J.W., J.Li, X.L., C.L., S.H., W.W., Y.Zhang, S.W., L.L., G.H., R.Zhao, K.Z. Rock scientific data analysis and geological interpretation: J.W., X.Z., Y.Zhao, and J.Liu. All authors reviewed and revised the manuscript.

## Competing interests

The authors declare no competing interests.

## Additional information

¹State Key Laboratory of Robotics and System, Harbin Institute of Technology, Harbin 150080, China. ²Beijing Aerospace Control Center, Beijing 100094, China. ³Center for Lunar and Planetary Sciences, Institute of Geochemistry, Chinese Academy of Science, Guiyang 550081, China. ⁴CAS Center for Excellence in Comparative Planetology, Hefei 230026, China. ⁵Key Laboratory of Science and Technology on Aerospace Flight Dynamics, Beijing 100094, China. ⁶Research Center for Planetary Science, College of Earth Science, Chengdu University of Technology, Chengdu 610059, China. ⁷Department of Aerospace Engineering, Ryerson University, Toronto, ON M5B 2K3, Canada. ⁸State Key Laboratory of Remote Sensing Science, Aerospace Information Research Institute, Chinese Academy of Sciences, Beijing 100101, China. ⁹These authors contributed equally: Liang Ding, Ruyi Zhou, Tianyi Yu. ✉e-mail: liangding@hit.edu.cn; gaohaibo@hit.edu.cn; lisirjian@163.com; liujianzhong@vip.gyig.ac.cn

