## [Peer Review File · Nature Communications]

Lunar rock investigation and tri-aspect characterization of lunar far side regolith by a digital twinEditorial Note: Parts of this peer review file have been redacted as indicated to avoid any copy right infringement.

REVIEWER COMMENTS

Reviewer #1 (Remarks to the Author):

In the manuscript, new observations by/of the Yutu-2 rover and a digital twin are used to estimate the regolith properties within Von Karman crater within the South Pole Aitken (SPA) basin and investigate the composition and origin of the fin-shaped rock near the Longji site.

As written, the manuscript does not provide sufficient details to evaluate the significance of the work and what impact the results from the described analyses will have on the field. Examples of what I mean:

(1) The title and text of the manuscript uses the descriptive term comprehensive when describing their analysis of regolith properties, however the authors do not describe what is meant by comprehensive (i.e., are the calculated parameters all the regolith properties you could calculate to fully characterize it?). Additionally, the manuscript does not include uncertainties on the modelled/calculated regolith property measurements to make it clear that the results are a significant contribution to the field. The internal friction angle was determined to be within 21.5 – 42.0 degrees and an associated cohesion of 520-3154 Pa, however it is only stated that these numbers make the farside regolith similar to the Apollo 12 soils. Without uncertainties, it is not clear if the manuscript includes significant results for publication.

(2) The inferred source of fin-shaped rock is based on the abundance of minerals derived from MGM and NBSR and comparing those results to derived abundances of minerals from Kaguya MI data. However, the manuscript does not include the uncertainties on these derived abundances and how Yutu-2 in-situ observations can be compared to orbital MI data. How unique is this approach in determining the origin of the rock. Additionally, the manuscript does not discuss the significance of knowing the origin of this rock (i.e., why does it matter that it came from Zhinyu crater rather than Finsen crater) or the significance of this being a lower crust or upper mantle rock.

(3) While equations are provided for the models in the Methods section, inputs into those models were not provided, nor were the assumptions described and how those assumptions and any uncertainties in the inputs play out to errors/uncertainties in the results presented in the paper. Thus, the significance of the results presented in the manuscript are unknown. Examples include:

(a) What is the uncertainty on the measurements of things like wheel sinkage from the imaging data? What is the spatial resolution of the imaging data?

(b) How are you measuring the size of regolith grains? Particularly at the micron scale?

(c) What scientific data, local terrain models and geometric and physical properties were used? What are their uncertainties?

(d) Some inputs seem to come from lunar simulants – how accurate are these simulants compared to real lunar regolith?

Additionally, much of the paper describes the traverse of the rover as well as previous results from the mission. While this is interesting, it isn't the most scientifically compelling. Another large portion of the paper describes the modeled behavior of the rover – again of limited scientific value as it is written.

Finally, the manuscript is not appropriately referenced as it is missing many of the recent publications on SPA in the past few years. It appears that they are using only references from their team and Chinese colleagues even though much work has been done by scientists from around the world.

Kerri Donaldson Hanna

Reviewer #2 (Remarks to the Author):

Using digital twin to infer geotechnical properties of lunar regolith is a very novel approach. In the past, I have seen using wheels to provide geotech properties based on DEM models - see for example Johnson et al., Discrete element method simulations of Mars Exploration Rover wheel performance, J of Terramechanics, Volume 62, December 2015, Pages 31-40

Page 3, Line 55: could you define 'lateral properties'?

Page 6, Line 122: could you define 'lateral shearing deformation modulus'?

Page 7, Line 129: could you define 'skid condition'?

Page 10, Line 180: spelling: variables and not varibles

Page 10, Line 193: see Johnson et al. above

Fig 3, 4 and others are too small. Pls increase the size.

Page 16, Line 294: How is Yutu-2 suspension different from NASA MER, MSL, and Mars2020? If it's the same, please state that. If its different - pls state how. Also - what was the reason for selecting this as opposed to other mobility architectures?

Supplementary Fig 3a - should be lateral and not laterial

Overall, this is an excellent paper!

Response to Referees

No.: NCOMMS-22-50761A-Z

Authors: Liang Ding^{1*†}, Ruyi Zhou^{1†}, Tianyi Yu^{2†}, Huaiguang Yang¹, Ximing He², Haibo Gao^{1*}, Juntao Wang^{3,4}, Ye Yuan¹, Jia Wang², Zhengyin Wang¹, Huanan Qi¹, Jian Li^{2*}, Wenhao Feng¹, Xin Li², Chuankai Liu^{2,5}, Shaojin Han², Xiaojia Zeng^{3,4}, Yu-Yan Sara Zhao^{6,4}, Guangjun Liu⁷, Wenhui Wan⁸, Yuedong Zhang², Saijin Wang², Lichun Li², Zongquan Deng¹, Jianzhong Liu^{3,4*}, Guolin Hu², Rui Zhao², Kuan Zhang²

Title: Digital twin enabled risky rock investigation and tri-aspect characterization of lunar farside regolith properties

We greatly appreciate the thorough review and constructive comments from the reviewers. All the comments have been addressed carefully with point-by-point response as listed in the following, and new descriptions and statements that are added into the revision in response to the reviewer's comments are in **red** color.

1. Response to the Comments of Review 1

In the manuscript, new observations by/of the Yutu-2 rover and a digital twin are used to estimate the regolith properties within Von Karman crater within the South Pole Aitken (SPA) basin and investigate the composition and origin of the fin-shaped rock near the Longji site. As written, the manuscript does not provide sufficient details to evaluate the significance of the work and what impact the results from the described analyses will have on the field. Examples of what I mean:

Response: Thank you for your comments. Detailed explanation of the uncertainty of input measurements and their effect on the parameter identification results have been added (*Response to Comment 1 (2)*). We added further analysis of the lunar regolith in terms of the bearing capacity and the bulk density, and more comparison on the shearing property of the lunar regolith at “Longji” site to that in Le Monnier crater has been supplemented to enrich the understanding of its properties (*Response to Comment 1 (3)*). Significance on the locomotion as well as interaction data present in this study have been emphasized in the revision (*Response to Comment 8*), and implications on parameter identification results have been discussed (*Response to Comment 1 (4)*).

Comment 1. 1) The title and text of the manuscript uses the descriptive term comprehensive when describing their analysis of regolith properties, however the authors do not describe what is meant by comprehensive (i.e., are the calculated parameters all the regolith properties you could calculate to fully characterize it?). **2)** Additionally, the manuscript does not include uncertainties on the modelled/calculated regolith property measurements to make it clear that the results are a significant contribution to the field. **3)** The internal friction angle was determined to be within 21.5 – 42.0 degrees and an associated cohesion of 520-3154 Pa, however it is only stated that these numbers make the farside regolith similar to the Apollo 12 soils. **4)** Without uncertainties, it is not clear if the manuscript includes significant results for publication.

Response to Comment 1 (1). The comprehensive here means the regolith

properties is analyzed and characterized in three aspects, which covers all the aspects (the bearing, shearing and lateral properties) concerned from terramechanics perspective. Admittedly, the word comprehensive may be too strong and not explicit enough to describe our analytical results, thus we substitute it with a milder and more concrete word **tri-aspect**, and change the title to *Digital twin-assisted risky rock investigation and tri-aspect characterization of lunar farside regolith properties* accordingly. We have revised the corresponding description in the revision as follows.

(Page 2, lines 3-4) **Leveraging digital twin technology, it comprehensively characterized the lunar regolith from three aspects and inferred the rock's origin.**

(Page 4, lines 10-12) In this work, we present the risky but successful venture to this peculiar rock enabled by digital twin with associated spectral investigation results, and the **tri-aspect** property identification of the farside lunar regolith achieved on special slipping and skidding states.

(Page 19, lines 18-20) The **tri-aspect** parameter identification results presented here serves as a foundational step for subsequent engineering verifications in the lunar farside, such as the assessments of landing feasibility, rover mobility, and sample excavation capabilities.

Response to Comment 1 (2). In terms of modelling, the terramechanics model we used for parameter identification is well established by considers multiple physical effects and well evaluated in previous study. For uncertainty on modelling, we have supplemented additional explanation on the accuracy of models in the *Methods* section (Page 28, lines 3-6).

These terramechanics models consider the slip sinkage phenomena and the wheel-lug effect, and could predict the normal force, resistance moment, and drawbar pull with higher accuracy compared to conventional terramechanics models⁷¹, and their modelling error is less than 10%⁴⁰ as illustrated in previous studies.

40. Ding, L. *et al.* Interaction mechanics model for rigid driving wheels of planetary rovers moving on sandy terrain with consideration of multiple physical effects. *J. Field Robot.* **32**, 827–859 (2015).

71. Wong, J. Y. *Theory of Ground Vehicles* (John Wiley & Sons, New Jersey, 2008).

For uncertainty on measurements, detailed description of uncertainties on measurements for regolith parameter identification have been added in the *Methods* section as follows (Page 29, lines 16-24; Page 30, lines 1-9).

As explicitly shown in Supplementary Fig. 11, inputs into these models for lunar regolith parameter estimation include forces (F_N , F_{DP} , F_L), wheel states (slip ratio s , sinkage z , side slip angle β), and wheel parameters (r , b , r_s). In this study, the normal force F_N , the drawbar pull F_{DP} and the lateral force F_L are solved through the constrained dynamics of a multi-rigid-body system with a floating base in the digital twin. The relative errors of estimated forces compared to experimental measurements are within 10%^{38,41}, as illustrated in previous studies. The wheel sinkage z is calculated by comparing the elevation difference of the track area and the surrounding surface plane in the reconstructed digital elevation model, which is reconstructed using stereo Navcam images in 1024×1024 pixels. The accuracy of the wheel sinkage measurement

is dependent on the accuracy of the single-site DEM, whose topographic data can reach the submillimeter level⁷³ (i.e., its error is within 1 mm). The wheel slip ratio s is solved with the actual travel distance of Yutu-2 estimated by visual localization and with the planned travel distance by rearranging Eq. (6). The accuracy of the slip ratio is dependent on the visual localization yielding an estimation error of 4%³⁹. The wheel side slip angle β is estimated using the posture of the rover, and the side slip angle along the forward traverse varied between 0.5° and 3° , whose accuracy depends on the visual positioning and its estimation error is 4%³⁹. The remaining rover wheel parameters r and h are constant depending on the rover configuration as listed in Supplementary Table 4, and the equivalent shearing radius r_s for Yutu-2's grouser wheel is calculated according to Eq. (2) with lug shearing coefficient of 0.65. The values or ranges of the inputs used in this study are in Supplementary Table 7.

38. Yang, H. *et al.* High-fidelity dynamic modeling and simulation of planetary rovers using single-input-multi-output joints with terrain property mapping. *IEEE Trans. Robot.* **38**, 3238–3258 (2022).
39. Liu, Z. *et al.* Landing site topographic mapping and rover localization for Chang'E-4 mission. *Sci. China Inf. Sci.* **63**, 140901 (2020).
41. Ishigami, G., Miwa, A., Nagatani, K., Yoshida, K. Terramechanics-based model for steering maneuver of planetary exploration rover on loose soil. *J. Field Robot.* **24**, 233–250 (2007).
73. Tang, Z. *et al.* Physical and mechanical characteristics of lunar soil at the Chang'E-4 landing site. *Geophys. Res. Lett.* **47**, e2020GL089499 (2020).

Supplementary Fig. 11. Regolith parameter estimation. Parameters in gray box are input of the framework, and they are fixed parameters set according to wheel parameters or obtained by the dynamic solution or external measurements. Equations in the orange boxes are key equations for parameter identification. Parameters in blue box are the output of the framework and are specific to terrain bearing, shearing and lateral parameters.

Supplementary Table 7 Inputs values of the wheel-terrain interaction models for parameter identification

Group	Variables (Unit)	Value
Forces	Normal force F_N (N)	32.02
	Drawbar pull F_{DP} (N)	-1.33
	Lateral force F_L (N)	-3.16
Wheel states	Slip ratio s	0 ~ -0.075

	Sinkage z (m)	0.005 ~ 0.015
	Side slip angle β ($^\circ$)	0.5 ~ 3
	Wheel radius r (m)	0.15
Wheel parameters	Wheel width b (m)	0.15
	Equivalent shearing radius of a wheel r_s (m)	0.1565

Based on the above uncertainty derived from different sources, we conducted an additional uncertainty analysis of identification results using Monte Carlo simulation. The results as well as analysis have been supplemented in the revision (Page 30, lines 10-24; Page 31, lines 1-10).

In this context, the primary source of error in identification outcomes stems from the measurement uncertainties associated with input variables, including slip ratio, sinkage, side slip angle and interaction forces. Due to the complexity and nonlinearity of our terramechanics models, getting an analytical uncertainty analysis of model output is a difficult task. Therefore, we use Monte Carlo simulations⁷⁴ to calculate the uncertainty of model outputs by considering input measurement uncertainties. These measurement errors are represented as normal distributions. The Monte Carlo simulation involves a sample size of 1024, which has been assessed as sufficient for obtaining reliable and robust results⁷⁵. Regarding bearing properties, when we assume a sinkage exponent (N) of 1.0, as the typical value from Apollo missions, the equivalent stiffness (K_s) ranged from 556.7 to 1443 kPa/m ^{N} for estimated sinkage of 8-15 mm. In other traverses, wheel sinkage may be smaller but generally not less than 5 mm. For very shallow sinkage, K_s approaches its upper limit with minimal adjustability. Consequently, for a sinkage of 5 mm, K_s was fixed at 1443 kPa/m ^{N} . To account for the measurement error associated with wheel sinkage, which was formulated as a normal distribution with a mean of 5 mm and a standard deviation of 0.33 mm (following the three-sigma rule), we conducted Monte Carlo simulations to estimate the corresponding distribution of output sinkage exponent N . The mean of the estimated N value is 0.87 with a standard deviation of 0.017 when only considering sinkage measurement uncertainties. When considering uncertainties in forces (F_N , F_{DP}) without accounting for the uncertainty of z (wheel sinkage), the standard deviation of the estimated N is 0.006, indicating that the predicted N is more sensitive to the wheel sinkage errors than to force errors in this case. By considering both uncertainties in forces and wheel sinkage, the standard deviation of the estimated N is estimated to be 0.017, as illustrated in Supplementary Fig. 13(a). For the largest sinkage of 15 mm, the standard deviation of the estimated K_s is approximately 25.8 kPa/m ^{N} (Supplementary Fig. 13(b)) when the sinkage exponent N is at the typical value of 1.0. Concerning shear properties, with a slip ratio of -0.075 and accounting for measurement errors on forces and slip ratio, the cohesion was determined to be 520 Pa (the typical values of the lunar regolith), and the internal friction angle φ was estimated to be 21.5 $^\circ$ with a standard deviation of 2.29 $^\circ$, as illustrated in Supplementary Fig. 13(c).

Supplementary Fig. 13 Uncertainty of the estimated mechanical properties using Monte Carlo simulations. **a**, Distribution of estimated sinkage exponents considering the measurement error of wheel sinkage and interaction forces. The results are conditioned on wheel sinkage of 5 mm with the standard deviation of 0.33 mm, and normal force of 32.027 N with 10% uncertainty, and drawbar pull of -1.33 N with 10% uncertainty. The mean of the sinkage exponent is 0.87 with standard deviation of 0.017. **b**, Distribution of estimated equivalent stiffness considering the measurement error of wheel sinkage and interaction forces. The results are conditioned on wheel sinkage of 15 mm with standard deviation of 0.33 mm, and the same force error as (a). **c**, Distribution of estimated internal friction angles considering the measurement error of slip ratio and force uncertainty. The results are conditioned on slip ratio of -0.075 with 4% uncertainty. The forces are in the same setting as (a).

74. Mooney, C. Z. *Monte carlo simulation* (New Delhi, London: SAGE Publications, 1997).

75. Zhou, R. Y. *et al.* Sensitivity analysis and dominant parameter estimation of wheel-terrain interaction model. *Acta Aeronautica et Astronautica Sinica* **42**, 524076 (2021).

Response to Comment 1 (3). For longitudinal shear properties, except for showing the similarity to regolith in the Apollo 12 mission, we emphasize the particularity in the increased cohesion for regolith at “Longji” site. Its cohesion was mostly larger than that measured at other sites, which was consistent with the greater soil cohesion phenomenon observed on the wheel. We ready have corresponding descriptions to the enhanced cohesion and further analyzed its implications for geological process they undergo in the manuscript as follows.

(Page 12, lines 23-24; Page 13, lines 1-6) Compared with the lunar regolith properties identified at other nearside landing sites⁴⁷ (Fig. 3c), the longitudinal shear property (characterized by cohesion and internal friction angle) of the lunar regolith at “Longji” site was closest to that estimated in the Apollo 12 mission using direct shear method⁴⁷. However, its cohesion was mostly larger than that measured at other sites, which was consistent with the great soil adhesion phenomenon observed on the wheel. The heightened degree of cohesion can be attributed to the increased presence of agglutinates within the lunar regolith, which constitute the principal product of space weathering⁴⁸. Consequently, this may suggest that the local lunar soil is subject to longer periods of exposure and more pronounced space weathering effects, leading to an enhanced level of regolith maturity⁴⁸.

48. Tang, Z. *et al.* Physical and mechanical characteristics of lunar soil at the Chang’E-4 landing site. *Geophys. Res. Lett.* **47**, e2020GL089499 (2020).

Further analyses on the regolith properties in terms of the bearing capacity have been supplemented to enrich the understanding of farside lunar regolith properties.

(Page 12, lines 6-10) With an average wheel sinkage of 8 mm along the rover's trace, the estimated bearing capacity reached approximately 4 kPa. This value was notably higher compared to the bearing capacity measured along the Lunohkod-1 traverse, where the uppermost layer at a depth of 1 cm exhibited a relatively smaller bearing capacity, falling within the range of 2-3 kPa¹⁹. This finding further substantiates the enhanced bearing strength of the lunar regolith at "Longji" site.

Response to Comment 1 (4). With all of these tri-aspect property identification results and added detailed uncertainty analysis, we think the results are a significant supplement for the lunar farside regolith property study, which is lack of data for delicate investigation/analysis. In addition, the locomotion experience and interaction data presented in this study is also valuable for further regolith mechanics studies. They are two of the main contributions of this paper to the field. We have emphasized the significance of the results for regolith mechanics studies and geological implications, and discussed their applications to the terramechanics and space exploration field as follows.

(Page 19, lines 6-9) The experience in locomotion and interaction yields valuable data for regolith mechanics studies, offering insights into how lunar regolith responds to a variety of static and dynamic loadings. This knowledge is essential for the development of terramechanics models that underpin interactions with rovers, landers, scoops, and other vehicles on the lunar surface¹⁹.

(Page 13, lines 3-6) The heightened degree of cohesion can be attributed to the increased presence of agglutinates within the lunar regolith, which constitute the principal product of space weathering⁴⁸. Consequently, this may suggest that the local lunar soil is subject to longer periods of exposure and more pronounced space weathering effects, leading to an enhanced level of regolith maturity⁴⁸.

(Page 19, lines 15-22) The in-situ identification based on wheel-terrain interaction models with the support of digital twin filled the gaps of the shear property estimation of the lunar farside regolith, and also provided a promising method to infer lateral properties of the regolith in extraterrestrial environments with initial parameters. The tri-aspect parameter identification results presented here serves as a foundational step for subsequent engineering verifications in the lunar farside, such as the assessments of landing feasibility, rover mobility, and sample excavation capabilities. Moreover, these findings may have implications for future in situ resource utilization (ISRU) technologies, where lunar regolith plays a pivotal role in base construction, mining, and resource extraction.

19. Slyuta, E. N. Physical and mechanical properties of the lunar soil (a review). *Solar Syst. Res.* **48**, 330-353 (2014).

48. Tang, Z. *et al.* Physical and mechanical characteristics of lunar soil at the Chang'E-4 landing site. *Geophys. Res. Lett.* **47**, e2020GL089499 (2020).

Comment 2. The inferred source of fin-shaped rock is based on the abundance of minerals derived from MGM and NBSR and comparing those results to derived abundances of minerals from Kaguya MI data. However, the manuscript does not include 1) the uncertainties on these derived abundances and 2) how Yutu-2 in-situ

observations can be compared to orbital MI data. **3)** How unique is this approach in determining the origin of the rock. **4)** Additionally, the manuscript does not discuss the significance of knowing the origin of this rock (i.e., why does it matter that it came from Zhinyu crater rather than Finsen crater) or the significance of this being a lower crust or upper mantle rock.

Response to Comment 2 (1). We have provided the error of the MGM estimation method in the original Fig. 4c, which is 0.267%. This is also the error in derived mineral abundances. We have added this point in the main text in the revised manuscript (Page 17, lines 13-15).

Note that the fitting error for the MGM method was 0.267% and since plagioclase was not included in the MGM methods, only relative abundance of mafic olivine, OPX and CPX were compared with the orbiter-derived data.

Response to Comment 2 (2)(3). It is a standard operation to trace the origins of rocks observed in situ to their source regions using the orbiter-derived data, particularly when there is a lack of in-situ observations or sample-returned analysis of the potential source regions. Yes, indeed, as the reviewer has pointed out, orbital-derived and in situ-derived spectral data are difficult to directly compare because of their difference in spatial scales, such as 100s or 1000s meter-scale versus centimeter-scale. However, if the mineralogy of the in-situ detection shows strong similarity to the spectra of in situ measurements, one can infer with greater confidence that the two spatial-scale observations are in agreement with each other²³; otherwise, additional data, both in situ and from orbital, would be necessary to conclude. In our case, the in-situ observation data are in good agreement with previous in-situ measurements of other rock targets, and it is also consistent with the derived mineralogical compositions of the Zhinyu crater. Therefore, here we conclude that the Fin-rock shares a similar origin with other previously studied rock targets along the traverse and ejecta from the Zhinyu Crater rather than the Finsen crater. We have discussed the limitation of such comparison with data derived from different spatial scale measurements in the relevant parts in the *In situ spectral investigation of the target rock section*.

(Page 17, lines 10-21) The abundances of minerals (i.e., olivine, OPX, and CPX), as indicated by the mosaic map created from topographically corrected Mineral Mapper reflectance data from the Kaguya MI, is similar to the modal mineralogy of ejecta from Zhinyu crater (Supplementary Table 3). Note that the fitting error for the MGM method was 0.267% and since plagioclase was not included in the MGM methods, only relative abundance of mafic olivine, OPX and CPX were compared with the orbiter-derived data. We understand that orbital-derived and in-situ-derived spectral data are difficult to compare due to the difference in spatial scales directly; however, because the in situ spectra suggest the fin-shaped rock is consistent with previously investigated rock targets along the traverse and in agreement with the derived mineralogical compositions of the Zhinyu crater, we infer that the target boulder belongs to the Zhinyu crater ejecta rather than those of the Finsen crater. Further investigations along the traverse of the Yutu-2 rover and potentially other sample-returned missions would be required to constrain the precise source regions of the local rocks.

23. Ma, P. *et al.* A plagioclase-rich rock measured by Yutu-2 rover in Von Karman crater

on the far side of the Moon. *Icarus* **350**, 113901 (2020).

Response to Comment 2 (4). We have expanded one paragraph in *Discussion* section to emphasize the scientific significance in determining the source region of the Fin-shaped rock (Page 20, lines 1-15).

The fin-shaped rock was homologous in spectral parameters to six rocks investigated previously in the Chang'E-4 mission and inferred to be sourced from the Zhinyu crater ejecta based on its relative abundance of 'OL-OPX-CPX' assemblages. This is the second large rock fragment inferred to be ejected from the Zhinyu crater along the traverse of the Yutu-2 rover. Zhinyu crater is a relatively young impact crater located about 30 km west of the landing site⁵⁸. The rover had closely investigated a previous rock fragment on the third lunar day and concluded it originated from the Zhinyu crater²³. Therefore, our discovery of the latest rock piece originated from Zhinyu crater supports that besides the majority of ejecta blanket from the Finsen crater to the CE-4 landing site, other source regions may also contribute to the local materials, suggesting a more complicated geological history in the local area. Meanwhile, the fin-shaped rock is also dominated by plagioclase, consistent with the previous suggestions that the CE-4 region mainly sampled the mare basalt instead of deeper mantle materials⁵⁸. These in-situ observations provide important ground truth for remote sensing investigation of the region and lay an essential foundation for reconstructing the thermal evolution history of the regional mare activities, and provide a better understanding of the geological context of SPA and the compositions of the lunar dichotomy at the farside, as well as the upcoming CE-6 sample-return mission⁵⁹.

23. Ma, P. *et al.* A plagioclase-rich rock measured by Yutu-2 rover in Von Karman crater on the far side of the Moon. *Icarus* **350**, 113901 (2020).

58. Gou, S. *et al.* Mare basalt flooding events surrounding Chang'e-4 landing site as revealed by Zhinyu crater ejecta. *Icarus* **360**, 114370 (2021).

59. Zeng, X. G. *et al.* Landing site of the Chang'e-6 lunar farside sample return mission from the Apollo basin. *Nat Astron*, 1-10 (2023).

Comment 3. While equations are provided for the models in the Methods section, 1) inputs into those models were not provided, 2) nor were the assumptions described 3) and how those assumptions and any uncertainties in the inputs play out to errors/uncertainties in the results presented in the paper. Thus, the significance of the results presented in the manuscript are unknown.

Response to Comment 3 (1). As mentioned in the *Response to Comment 1 (2)*, additional explanation regarding inputs into models with associated uncertainties have been added in the revision.

Response to Comment 3 (2) Thanks for the valuable suggestion. We have added the model assumption at the end of the corresponding *Method* section as follows (Page 28, lines 6-9).

The bearing model (Eq. (1a)) is adopted from Bekker model⁷² with semi-empirical equations, thus it is based on the assumption that the wheel slip ratio is no larger than 0.6⁶⁷. For high slippery ($s > 0.6$) cases, the accuracy of the above semi-empirical terramechanics models will get significantly affected, and cannot reach a reasonable

agreement with the wheel mobility performance.

67. Ding, L. Wheel-Soil Interaction Terramechanics for Lunar/Planetary Exploration Rovers: Modeling and Application. PhD thesis, Harbin Institute of Technology (2009).
72. Bekker, M. G. *Off-the-road-locomotion* (Ann Arbor, MI: The University of Michigan Press, 1960).

Response to Comment 3 (3) For assumptions, additional explanation on the model applicability in our case has been added in the revision (Page 28, lines 9-14).

In this study, the rover wheel work on skid condition during the outbound traverse, and most the wheel slip ratios are between 0 and -0.075 and no less than -0.1, satisfying the slip ratio assumption and can be applied to parameter identification. When the slip ratio is over 0.6, the parameter identification process can still obtain a set of parameters in the sense of fitting, but the model is no longer in high accuracy in such condition, thus the accuracy of the identified parameter is also affected.

For inputs uncertainties, as mentioned in the *Response to Comment 1 (2)*, we have added additional uncertainty analysis on identification results based on Monte Carlo simulations.

Examples include:

Comment 4 (a) What is the uncertainty on the measurements of things like wheel sinkage from the imaging data? What is the spatial resolution of the imaging data?

Response to Comment 4. The accuracy of the wheel sinkage measurement is dependent on the accuracy of the single-site digital elevation model (DEM) reconstructed from Navcam images, whose topographic data reaches the submillimeter level. The spatial resolution of the Navcam imaging data is 1024×1024 pixels. Except for the wheel sinkage z , other measurements involved in regolith parameter identification are wheel slip ratio s and side slip angle β . The accuracy of wheel slip ratio s is dependent on visual localization yielding an estimation error of 4%³⁹. The side slip angle β is estimated with the posture of the rover, whose estimation error is 4%³⁹. Details about measurement uncertainties for regolith parameter identification have been added in the ‘**Lunar regolith parameter estimation at the “Longji” site**’ subsection as follows (Page 29, lines 21-24; Page 30, lines 1-6).

The wheel sinkage z is calculated by comparing the elevation difference of the track area and the surrounding surface plane in the reconstructed digital elevation model, which is reconstructed using stereo Navcam images in 1024×1024 pixels. The accuracy of the wheel sinkage measurement is dependent on the accuracy of the single-site DEM, whose topographic data can reach the submillimeter level⁷³ (i.e., its error is within 1 mm). The wheel slip ratio s is solved with the actual travel distance of Yutu-2 estimated by visual localization and with the planned travel distance by rearranging Eq. (6). The accuracy of the slip ratio is dependent on the visual localization yielding an estimation error of 4%³⁹. The wheel side slip angle β is estimated using the posture of the rover, and the side slip angle along the forward traverse varied between 0.5° and 3°, whose accuracy depends on the visual positioning and its estimation error is 4%³⁹.

39. Liu, Z. *et al.* Landing site topographic mapping and rover localization for Chang’E-

- 4 mission. *Sci. China Inf. Sci.* **63**, 140901 (2020).
73. Tang, Z. *et al.* Physical and mechanical characteristics of lunar soil at the Chang'E-4 landing site. *Geophys. Res. Lett.* **47**, e2020GL089499 (2020).

Measurements used for rock identification include the visible and near-infrared (VNIR) hyperspectral images and single-pixel short-wave infrared (SWIR) spectra from the VNIS instrument. The measurement uncertainties are 5% and 7% for VIS/NIR and SWIR, respectively. We further claimed the uncertainty of the scientific data used in this study at the end of the “**Instruments and data description**” section as follows (Page 23, lines 11-12).

The measurement uncertainties are 5% and 7% for VIS/NIR and SWIR, respectively⁶⁵.

65. Li, C. *et al.* Chang'E-4 initial spectroscopic identification of lunar far-side mantle-derived materials. *Nature* **569**, 378–382 (2019).

Comment 5. How are you measuring the size of regolith grains? Particularly at the micron scale?

Response to Comment 5. The size of the regolith grains was estimated from the image. We have removed “micro- to millimeter-sized” from the original manuscript due to the lack of a reference scale for this image. A fine-grain regolith was a consensus for the grains attached to the rock surface.

Comment 6. What 1) scientific data, 2) local terrain models and geometric and physical properties were used? What are their uncertainties?

Response to Comment 6 (1). The scientific data used in this study includes Pancam images, the visible and near-infrared (VNIR) hyperspectral images and single-pixel short wave infrared (SWIR) spectra from the VNIS instrument. Images from Pancam are 2352×1728 pixels in color mode or 1176×864 pixels in panchromatic mode as illustrated in Supplementary Table 5. The visible and near-infrared (VNIR) hyperspectral images are 256×256 pixels as illustrated in Supplementary Table 6. The measurement uncertainties are 5% and 7% for VIS/NIR and SWIR, respectively. We further claimed the scientific data used in this study with its measurement uncertainty in the *Instruments and data description* section as follows.

(Page 22, lines 17-19) The data used in this study include images from a Pancam, Navcam, and two HazCams, alongside the visible and near-infrared (VNIR) hyperspectral images and single-pixel short-wave infrared (SWIR) spectra from the VNIS instrument.

(Page 22, lines 20-22) Pancam is one of the scientific payloads used primarily for high-resolution mapping and localization. It is mounted on the mast of the Yutu-2 rover, consisting of two optical systems of identical functions, performances, and interfaces, parameterized as Supplementary Table 5.

(Page 23, lines 6-12) The VNIS^{65,66} is composed of a complementary metal-oxide-semiconductor (CMOS) image with 256 by 256 pixels, a SWIR spectrometer, and a white panel for calibration and dust-proofing. It is assembled on the anterior of

the rover to detect the composition of the lunar surface materials at a fixed 45° zenith angle at a height of 0.69 m. Its spectral wavelength ranges are 450-945 nm and 900-2395 nm with a default sampling interval of 5 nm. For the SWIR spectrometer, its field of view is a circular area in the CMOS image centered at (96, 128) with a 54 pixel radius (Supplementary Table 6). The measurement uncertainties are 5% and 7% for VIS/NIR and SWIR, respectively⁶⁵.

65. Li, C. *et al.* Chang'E-4 initial spectroscopic identification of lunar far-side mantle-derived materials. *Nature* **569**, 378–382 (2019).

66. He, Z. *et al.* Spectrometers based on acousto-optic tunable filters for in-situ lunar surface measurement. *J. Appl. Remote Sens.* **13**, 027502 (2019).

Response to Comment 6 (2). The local terrain model with geometric and physical properties used in this study is a digital elevation model (DEM) with physical properties named DEMP² proposed from ref. 38. Additional explanation of the local terrain model with geometric and physical properties with modeling accuracy and parameter settings have been added in the revision (Page 33, lines 12-22).

The local terrain model with geometric and physical properties used in this study is a digital elevation model (DEM) with physical properties named DEMP² proposed from ref. 38. As shown in Supplementary Fig. 12, the DEMP² is separated into discrete nodes and triangle elements, and each node has its own physical properties. The coordinates of terrain nodes are given as $(x, y, z, K_s, N, c, \varphi, K_x, \varphi_y, K_y)$, in which (x, y, z) represent 3-dimensional position of the node in inertial frame, i.e., geometric properties, and $(K_s, N, c, \varphi, K_x, \varphi_y, K_y)$ are physical properties of the node. The local terrain model of the “Longji” site was reconstructed with 0.02m spatial resolution based on Pancam stereo images³⁹, and its raw geometric properties represented in elevation map is shown in Supplementary Fig. 1a. The topographic data from the single-site DEM reaches the submillimeter level⁷³. In the simulation, the physical properties to be identified are under determined, and set in a wide range of parameters to cover most cases, while non-dominant physical parameters are set as the typical value of the lunar regolith from Apollo data⁴⁷, as listed in Supplementary Table 8.

Supplementary Fig. 12 Description of the DEMP² used in the dynamic simulation

Table S2 Nomenclature

Nomenclature	Meaning
K_s	equivalent stiffness modulus of the soil
N	variable sinkage exponent of the wheel-terrain interaction
c	cohesion of the soil

φ	internal friction angle of the soil
K_x	longitudinal shearing deformation modulus of the soil
φ_y	external friction angle of the soil
K_y	lateral shearing deformation modulus of the soil

Supplementary Table 8. Setting of the terrain mechanical parameters in the simulation

Phase	Number of groups	$K_s(\text{kPa}/\text{m}^N)$	N	$c(\text{Pa})$	$\varphi(^{\circ})$	$K_x(\text{mm})$	$\varphi_y(^{\circ})$	$K_y(\text{mm})$
Outboard	7	827	1.0	520	25-55	17.8	$0.3 \times \varphi$	40
journey	7	827	1.0	520	40	17.8	$0.3 \times \varphi$	15-45
Return	1(unrectified)	827	1.0	520	42	17.8	12.6	20
journey	1(rectified)	827	0.9	520	38.95	17.8	12.6	40

38. Yang, H. *et al.* High-fidelity dynamic modeling and simulation of planetary rovers using single-input-multi-output joints with terrain property mapping. *IEEE Trans. Robot.* **38**, 3238–3258 (2022).
39. Liu, Z. *et al.* Landing site topographic mapping and rover localization for Chang'E-4 mission. *Sci. China Inf. Sci.* **63**, 140901 (2020).
47. French, B. M., Heiken, G., Vaniman, D., Schmitt, H. H., & Schmitt, J. *Lunar Sourcebook A Users Guide to the Moon* (Cambridge Univ. Press, 1991).
73. Tang, Z. *et al.* Physical and mechanical characteristics of lunar soil at the Chang'E-4 landing site. *Geophys. Res. Lett.* **47**, e2020GL089499 (2020).

Comment 7. (d) Some inputs seem to come from lunar simulants – how accurate are these simulants compared to real lunar regolith?

Response to Comment 7. Due to lack of lateral property data measured with real lunar regolith, we had to compare our identification results with the released lateral parameters of lunar simulants. These two kinds of simulant samples that we used for lateral parameter comparison are derived from ref. 41 and ref. 82. The simulants from ref. 41 is a lunar regolith simulant which is equivalent to FJS-1⁸¹, whose material components and mechanical characteristics are similar to those of the real lunar soil⁴⁷, as listed in Supplementary Table S1 and 11. The simulants from ref. 82 is made from standard commercial experimental sand, numbered HIT-LSS2, whose particle size is set to be uniform so that the repeatability of the sand is secured. Its physical and mechanical property parameters are listed as Supplementary Table 12.

We have added additional explanation of the accuracy of these simulants compared to real lunar regolith in the revision as follows (Page 15, lines 11-16).

The lunar simulant is equivalent to FJS-1⁸¹, whose material components and mechanical characteristics (Supplementary Table 11) are similar to those of the real lunar soil, as reported in ref. 47. Another simulant (Supplementary Table 12) is standard commercial experimental sand, numbered HIT-LSS2, whose particle size is set to be uniform for repeatability of terramechanics experiments. The lateral parameters of the lunar simulant and the standard commercial experimental sand are from ref. 41 and 82.

Supplementary Table S1 Major chemical compositions of Apollo samples and simulants⁸¹

Oxide	Lunar Samples (Mean)									Simulants
	Apollo Missions						Luna Missions			
	11	12	14	15	16	17	16	20	24	
SiO ₂	42.2	46.3	48.1	46.9	45.0	43.2	41.7	45.1	43.9	49.1
TiO ₂	7.8	3.0	1.7	1.4	0.54	4.2	3.4	0.55	1.3	1.9

Al ₂ O ₃	13.6	12.9	17.4	14.6	27.3	17.1	15.3	22.3	12.5	16.2
Cr ₂ O ₃	0.30	0.34	0.23	0.36	0.33	0.33	0.28	-	0.32	-
FeO	15.3	15.1	10.4	14.3	5.1	12.2	16.7	7.0	19.8	8.3
Fe ₂ O ₃	-	-	-	-	-	-	-	-	-	4.8
MnO	0.20	0.22	0.14	0.19	0.30	0.17	0.23	0.13	0.25	0.19
MgO	7.8	9.3	9.4	11.5	5.7	10.4	8.8	9.8	9.4	3.8
CaO	11.9	10.7	10.7	10.8	15.7	11.8	12.5	15.1	12.3	9.1
Na ₂ O	0.47	0.54	0.70	0.39	0.46	0.40	0.34	0.50	0.31	2.8
K ₂ O	0.16	0.31	0.55	0.21	0.17	0.13	0.10	0.10	0.04	1.0
P ₂ O ₅	0.05	0.4	0.51	0.18	0.11	0.12	0.12	0.16	0.11	0.44
S	0.12	-	-	0.06	0.07	0.09	0.21	0.08	0.14	-
H ₂ O	-	-	-	-	-	-	-	-	-	0.43
Total	99.9	99.6	99.8	100.8	100.8	100.5	99.7	100.8	100.4	98.1

Supplementary Table 12 Mechanical properties of Apollo samples and simulants⁸¹

	Bulk Density (g/cm ³)	Particle Specific Gravity	Shear Strength	
			Cohesion c (kPa)	Friction Angle φ (°)
Lunar Soil (Averaged)	1.53 – 1.63 (Near Surface)	2.3 - 3.2	0.1 - 1	30 – 50
FJS-1	1.55	2.94	8	37.2
Lunar soil simulant ⁴²	1.6	-	0.8	37.2

Supplementary Table 13 The physical and mechanical parameters of the lunar soil simulant and HIT-LSS2^{82, 85}

	ρ (Kg/m ³)	k_c (kPa/m ^{n-1})	k_φ (kPa/m ^{n})	n	c (kPa)	φ (°)	K_x (m)	K_y (m)
Lunar soil simulant ⁴²	1.600	1.71	4754.67	1.00	0.8	37.20	0.043 β + 0.036	0.020 β + 0.013
HIT- LSS2 ^{82,85}	1.600	0	1043.0	0.87	0.46	38.1	0.005503 β + 0.008302	0.045 β + 0.01201

Note: ρ is bulk density, k_c is cohesion modulus of the soil, k_φ is the frictional modulus of the soil, n is the sinkage exponent, c is the cohesion of the soil, φ is the internal friction angle of the soil, K_x is the longitudinal shearing deformation modulus of the soil, K_y is the lateral shearing deformation modulus of the soil.

41. Ishigami, G., Miwa, A., Nagatani, K., Yoshida, K. Terramechanics-based model for steering maneuver of planetary exploration rover on loose soil. *J. Field Robot.* **24**, 233–250 (2007).
47. French, B. M., Heiken, G., Vaniman, D., Schmitt, H. H., & Schmitt, J. *Lunar Sourcebook A Users Guide to the Moon* (Cambridge Univ. Press, 1991).
81. Kanamori, H., Udagawa, S., Yoshida, T., Matsumoto, S., Takagi, K., Properties of lunar soil simulant manufactured in Japan. In *the 6th International Conference and Exposition on Engineering, Construction, and Operations in Space* (1998).
82. Li, J. Research on wheel soil interaction mechanics for planetary exploration rovers under cornering and slip conditions. Master thesis, Harbin Institute of Technology (2017).
85. Ding, L. *et al.* Experimental Study and Analysis of the Wheels' Steering Mechanics for Planetary Exploration WMRs Moving on Deformable Terrain. *Int. J. Robot. Res.* **32**, 712–743 (2013).

Comment 8. 1) Additionally, much of the paper describes the traverse of the rover

as well as previous results from the mission. While this is interesting, it isn't the most scientifically compelling. 2) Another large portion of the paper describes the modeled behavior of the rover – again of limited scientific value as it is written.

Response to Comment 8 (1). Thank you for the comments. We have further emphasized the significance on rover traverse analysis and discussed its impact on other studies in the revision.

(Page 5, lines 7-16) Such robotic geological exploration to desired scientific interesting target relies significantly on the successful traversal of rovers across diverse extraterrestrial terrains. However, from the locomotion perspective, this potential traverse was full of mobility hazards and uncertainties. On the one hand, the Yutu-2 rover was likely to suffer highly risky wheel skidding³⁰ when moving downwards on such a slope. Furthermore, uncontrollable lateral slippage would inevitably occur with broken nonholonomic constraints of the rover during the traverse, thus the path-following accuracy of Yutu-2 is hard to guarantee. Locomotion failures on harsh or slippery terrains have ever brought serious consequences, such as reduced tractive performance, deviation from planned trajectories, and in the worst-case scenario, becoming immobilized and permanently trapped, exemplified on previous missions on the Moon (like Lunar Roving Vehicle (LRV)³², Luna 21 and Lunokhod 2³³) and Mars exploration missions^{34,35}.

(Page 5, line 24; Page 6, lines 1-4) To keep the mission continue, movements on harsh terrains with slipping risk must be justified to remain within the acceptable bounds of the rover's mobility safety margin prior to execution.

(Page 19, lines 6-9) The experience in locomotion and interaction yields valuable data for regolith mechanics studies, offering insights into how lunar regolith responds to a variety of static and dynamic loadings. This knowledge is essential for the development of terramechanics models that underpin interactions with rovers, landers, scoops, and other vehicles on the lunar surface¹⁹.

19. Slyuta, E. N. Physical and mechanical properties of the lunar soil (a review). *Solar Syst. Res.* **48**, 330-353 (2014).
32. Costes, N. C., Farmer, J. E., George, E. B. Mobility performance of the lunar roving vehicle: terrestrial studies - Apollo 15 results. NASA Marshall Space Flight Center, NASA Technical Report TR R-401; 1972.
33. Carrier, III W. Soviet Rover Systems. In Proceedings of the *Space Programs and Technology Conference*, Huntsville, USA, 1992.
34. Arvidson, R. *et al.* Mars science laboratory curiosity rover megaripple crossings up to Sol 710 in Gale Crater. *J Field Robot.* **34**, 495–518 (2017).
35. Wilcox B, Nguyen T. Sojourner on Mars and Lessons Learned for Future Planetary Rovers. In Proceedings of the *SAE International Conference on Environmental Systems*, Danvers, MA, 1998.

Response to Comment 8 (2). Additional statistical analysis on the behavior modelling error has been added to strengthen the prediction results; the significance of accurate behavior modelling for rover mobility prediction and mission success has been further stressed; and the hidden philosophy of digital twin exemplified by behavior

modelling and its insight into broad engineering and scientific problems have been discussed in the revision.

(Page 9, lines 13-16) The lateral model employed for representing wheel side force underwent rigorous validation to enhance its predictive performance for steering trajectories, achieving a final state error of less than 15%⁴¹. Additionally, this model exhibits a high degree of accuracy in estimating the rover's orientation⁴¹.

(Page 9, lines 21-23) The small discrepancy between the predicted state and the real-world state underscores the robustness and efficacy of the digital twin system, uniquely positioned to enhance decision support and elevate the mission success rate.

(Page 19, lines 4-6) Precise locomotion simulation and thorough pre-traverse evaluation significantly mitigate the risks of mission failure by meticulously modeling wheel-terrain interactions and leveraging real-world data.

(Page 14, lines 4-9) The elaborate wheel-regolith interaction models, adaptable to various motion conditions, constitute the cornerstone for predicting the rover's realistic mobility behavior within the digital twin system, while the real-world data is used to update the models towards higher fidelity. The high integration of models and data within the digital system exemplified by behavior prediction demonstrate the system's remarkable capability to bridge the gap between simulation and reality, with broad implications for other sim2real challenges.

41. Ishigami, G., Miwa, A., Nagatani, K., Yoshida, K. Terramechanics-based model for steering maneuver of planetary exploration rover on loose soil. *J. Field Robot.* **24**, 233–250 (2007).

Comment 9. Finally, the manuscript is not appropriately referenced as it is missing many of the recent publications on SPA in the past few years. It appears that they are using only references from their team and Chinese colleagues even though much work has been done by scientists from around the world.

Response to Comment 9. Thank you for this valuable feedback. In this revision, we have prepared a separate paragraph reviewing the SPA studies conducted by international contributors besides the Yutu-2 team and included more literature in this paragraph. In the first paragraph of the *Introduction*, we have also added two papers led by the international groups to provide a more balanced overview of the study context.

(Page 2, lines 18-22) As the oldest and largest impact basin on the Moon, the South Pole-Aitken (SPA) basin is one of the most appealing farside places that is supposed to have exposed the lunar lower crust and probably upper mantle materials^{3,4}, and promising to reveal the indeterminate evolution of the early Moon with oldest mare basalts ever detected^{5,6}.

(Page 3, lines 3-16) Estimates of lunar crustal thickness obtained from the GRAIL mission corroborate the notion that the SPA impact event likely excavated materials deep into the mantle⁸, and there presents a large excess of mass in the lunar mantle under the SPA⁹. Additional evidence from remote sensing, impact modeling, and geological analyses indicates that the SPA impact ejected ilmenite-bearing cumulates (IBCs) and KREEP-bearing rocks from the uppermost mantle^{10,11}. Continuous spectral reflectance data acquired by the Spectral Profiler instrument aboard the lunar explorer

SELENE/Kaguya reveal enriched FeO contents in the central depression of the SPA, indicating the presence of mafic materials such as impact melt breccia¹². Orbital spectral observations of the materials within the SPA strongly suggest the excavations of the lunar mantle; however, low-Ca pyroxene (LCP)-rich rocks are more numerous and more widely distributed than olivine-rich rocks, dominating spectral signatures of the mantle-derived SPA impact melts^{4,13-17}. Whether the sparse distribution of olivine-rich materials within the SPA is due to the lack of in-situ measurements with high resolution, or is it indicative of the layered structure of the lunar mantle, a horizontal heterogeneity in mantle composition, or the impact origin of the basin, remains open questions. Notably, the definitive identification of mantle materials, whether on the lunar surface or in the analysis of returned samples, remains elusive.

4. Yamamoto, S. *et al.* Lunar Mantle Composition Based on Spectral and Geologic Analysis of Low-Ca Pyroxene- and Olivine-Rich Rocks Exposed on the Lunar Surface. *J Geophys Res-Planet* **128**, e2023JE007817 (2023).
6. Ivanov, M. A. *et al.* Geologic History of the Northern Portion of the South Pole-Aitken Basin on the Moon. *J Geophys Res-Planet* **123**, 2585-2612 (2018).
8. Wieczorek M A, *et al.* The crust of the Moon as seen by GRAIL. *Science*, **339**, 671-675 (2013).
9. James, P. B. *et al.* Deep Structure of the Lunar South Pole-Aitken Basin. *Geophys Res Lett* **46**, 5100-5106 (2019).
10. Moriarty, III D. P., *et al.* Evidence for a stratified upper mantle preserved within the South Pole-Aitken basin, *J. Geophys. Res.-Planets* **126**, e2020JE006589 (2021).
11. Moriarty III, D. P., *et al.* The search for lunar mantle rocks exposed on the surface of the Moon. *Nat. Commun.* **12**, 4659 (2021).
12. Naito, M. *et al.* Iron distribution of the Moon observed by the Kaguya gamma-ray spectrometer: Geological implications for the South Pole-Aitken basin, the Orientale basin, and the Tycho crater. *Icarus* **310**, 21-31 (2018).
13. Melosh, H. J. *et al.* South Pole–Aitken basin ejecta reveal the Moon’s upper mantle. *Geology* **45**, 1063–1066 (2017).
14. Nakamura, R. *et al.* Ultramafic impact melt sheet beneath the South Pole–Aitken basin on the Moon. *Geophys. Res. Lett.* **36**, L22202 (2009).
15. Moriarty, D. P., Pieters, C. M. & Isaacson, P. J. Compositional heterogeneity of central peaks within the South Pole-Aitken Basin. *J. Geophys. Res. Planets* **118**, 2310–2322 (2013).
16. Ohtake, M. *et al.* Geologic structure generated by large-impact basin formation observed at the South Pole-Aitken basin on the Moon. *Geophys. Res. Lett.* **41**, 13161 (2014).
17. Moriarty, D. P. & Pieters, C. M. The character of South Pole-Aitken Basin: patterns of surface and subsurface composition. *J. Geophys. Res. Planets* **123**, 729–747 (2018).

2. Response to the Comments of Review 2

Comment 1. Using digital twin to infer geotechnical properties of lunar regolith is a very novel approach. In the past, I have seen using wheels to provide geotech properties based on DEM models - see for example Johnson et al., Discrete element method simulations of Mars Exploration Rover wheel performance, *J of Terramechanics*, Volume 62, December 2015, Pages 31-40

Response to Comment 1. We greatly appreciate the reviewer's positive comments and constructive suggestions to help us improve the manuscript. Thanks for sharing the reference using discrete element method for micro-scale properties study, which was a great supplement for the microscopic properties investigated in this work. We have added it in the discussion as follows.

(Page 20, lines 19-21) **In addition to the macroscopic properties of the lunar soil, its microscopic properties are also worth an in-depth examination, which can be achieved through discrete element-based wheel-regolith interaction simulations⁶⁰.**

60. Johnson, J. *et al.* Discrete element method simulations of Mars Exploration Rover wheel performance, *J. Terramech.* **62**, 31-40 (2015).

Comment 2. Page 3, Line 55: could you define 'lateral properties'?

Response to Comment 2. The lateral properties of the lunar regolith characterize the lateral force on the wheel during the wheel-terrain interaction when the wheel has a lateral travelling velocity v_y . The properties are parameterized by external friction angle ϕ_y and the lateral shearing deformation modulus of the soil K_y , except for the cohesion c .

When the forward velocity of the wheel v deviates from its heading direction v_x due to the existence of lateral travelling velocity v_y , the wheel is subjected to an additional lateral resistance force F_L acting along the lateral direction of the wheel to resist this lateral movement. The lateral resistance force F_L is integrated by the lateral shear stress τ_y , and the soil lateral properties are parameters characterizing the lateral force or lateral shear stress of the wheel during the wheel-terrain interaction. According to the ref. 41, the soil lateral properties are parameterized by the cohesion c , external friction angle ϕ_y and the lateral shearing deformation modulus of the soil K_y in the lateral shear stress model, similar to the representation of the longitudinal shear properties characterized by cohesion c , internal friction angle ϕ and the longitudinal shearing deformation modulus K_x . The external friction angle ϕ_y represents the roughness between the wheel surface and the soil, and the lateral shearing deformation modulus of the soil K_y represents the tangential shear strength of the soil. We explained the definition of lateral properties in the revision (Page 13, lines 8-13).

The lateral properties of the lunar regolith characterize the lateral force on the wheel during the wheel-terrain interaction when the wheel has a lateral travelling velocity v_y . The properties are parameterized by the external friction angle ϕ_y and the lateral shearing deformation modulus of the soil K_y . The external friction angle ϕ_y represents the roughness between the wheel surface and the regolith, and the lateral

shearing deformation modulus of the soil K_y represents the tangential shear strength of the regolith.

Additional summation of all three aspects of soil properties has been added at the end of the *wheel-terrain interaction modelling* section in Methods (Page 27, line 16-19; Page 28, lines 1-2).

In our models, the terrain mechanical parameters are characterized in bearing, longitudinal shearing and lateral shearing three aspects. The bearing properties are related to the bearing strength of the soil represented in K_s and N ; while the shearing properties regarding soil shear strength are characterized in the longitudinal and tangential directions. They are symbolized by c , φ , K_x and φ_y , K_y , respectively. The external friction angle φ_y represents the roughness between the wheel surface and the soil, and the lateral shearing deformation modulus K_y represents the tangential shear strength of the soil.

41. Ishigami, G., Miwa, A., Nagatani, K., Yoshida, K. Terramechanics-based model for steering maneuver of planetary exploration rover on loose soil. *J. Field Robot.* **24**, 233–250 (2007).

Comment 3. Page 6, Line 122: could you define 'lateral shearing deformation modulus'?

Response to Comment 3. The definition of 'lateral shearing deformation modulus' is the soil deformation modulus in lateral direction⁴². The soil shear deformation modulus K is equivalent to a time constant of the stress-deformation curve, and the value of K is defined by the interaction mechanics between wheel and soil. The shear deformation modulus K can be divided into longitudinal and lateral directions symbolized by K_x and K_y respectively, and they are empirically estimated as functions of slip angle β .

We added the corresponding definition where the lateral shearing deformation modulus appeared for the first time in the text as follows (Page 8, line 18-21) and the nomenclature sheet in the supplementary as well.

A wide range of parameters for regolith was set in the wheel-terrain interaction model to cover most cases, e.g., the internal friction angle φ varied from 25° to 55°, and the lateral shearing deformation modulus K_y (the soil deformation modulus in lateral direction⁴²) was within the range of 15 to 45 mm.

42. G. Ishigami, Terramechanics-based Analysis and Control for Lunar/Planetary Exploration Robots. PhD thesis, Tohoku University (2008).

Comment 4. Page 7, Line 129: could you define 'skid condition'?

Response to Comment 4. The skid condition is the negative slip condition when the circumference velocity of wheel is smaller than the traveling velocity of wheel. We have added the definition of the 'skid condition' in the revision (Page 9, lines 2-5).

During the outbound journey, simulation results (Supplementary Fig. 5) show that rover wheels work mostly in skid condition (a negative slip condition when the circumference velocity of wheel is smaller than the traveling velocity of wheel), and suffer lateral slippage to a certain extent.

In addition, we have added the description of ‘slip condition’ as well to distinguish the difference between those two conditions (Page 25, lines 12-13).

If $s > 0$, the wheel is in slip condition; if $s = 0$, the wheel rolls without slipping and skidding; and if $s < 0$, the wheel is in skid condition.

Comment 5. Page 10, Line 180: spelling: variables and not variables

Response to Comment 5. Revised.

Comment 6. Page 10, Line 193: see Johnson et al. above

Response to Comment 6. Thanks for sharing the reference. Johnson *et al.* leveraged the three-dimensional discrete element method (DEM) simulations of wheel drawbar pull and sinkage to determine the particle packing density (0.62) and the interparticle friction coefficient (0.8) with MIT MER wheel performance testing data. These geotechnical properties investigated in this reference are the micro-scale properties of the soil related to the interparticle interaction only under longitudinal motion, while the soil properties studies in this work are macro-scale properties concerned in the wheel-soil interaction under both longitudinal and steering motion. The micro-scale properties of the soil presented in the reference is a great complement to our work, but there is no lateral property of the regolith illustrated and no in-situ results. Therefore, to our best knowledge, our work is still the first to investigate the lateral property of the lunar regolith via in-situ measurements.

We have added the suggested reference as future research direction towards micro-scale properties using the discrete element method in the discussion.

(Page 20, lines 19-21) **In addition to the macroscopic properties of the lunar soil, its microscopic properties are also worth an in-depth examination, which can be achieved through discrete element-based wheel-regolith interaction simulations⁶⁰.**

60. Johnson, J. *et al.* Discrete element method simulations of Mars Exploration Rover wheel performance, *J. Terramech.* **62**, 31-40 (2015).

Comment 7. Fig 3, 4 and others are too small. Pls increase the size.

Response to Comment 7. Thank you for this point. We have enlarged and reformatted Fig. 3, 4 and others (like Supplementary Fig. 10) according to the journal image formatting requirements.

Comment 8. 1) Page 16, Line 294: How is Yutu-2 suspension different from NASA MER, MSL, and Mars2020? If it's the same, please state that. If it's different - pls state how. **2)** Also - what was the reason for selecting this as opposed to other mobility architectures?

Response to Comment 8 (1). Yutu-2 is similar in suspension configuration to NASA MER, MSL and Mars 2020. All of them use the rocker-bogie suspension that connects the six wheels to the body of the rover with three main components: differential, rocker and bogie. The differential mechanism of Yutu-2 is inside the body, which is similar to the counterpart of MER, but different from MSL and Mars2020, whose differential mechanism are outside of the body connecting to the left and right

rockers and to the rover body by a pivot in the center of the rover's top deck. Except for the suspension system, these rovers are different in wheel designs as shown in Supplementary Fig. 10.

We have added statement about the same suspension configuration and different wheel designs to NASA series of rovers with additional references as follows.

(Page 21, lines 19-23; Page 22, lines 1-3) **The Yutu-2 rover (Supplementary Fig. 2b) is a six-wheeled robot designed in a rocker-bogie suspension, which is similar to NASA's series of Mars rovers (Spirit, Opportunity⁶⁴, Curiosity (<https://mars.nasa.gov/msl/spacecraft/rover/wheels/>), and Perseverance (<https://mars.nasa.gov/mars2020/spacecraft/rover/wheels/>)) in suspension configuration. An internal differential mechanism connects the left and right rocker bogies assemblies to the rest of the vehicle, which is different from rovers in Mars Science Laboratory (MSL) and Mars 2020 missions, whose differential mechanism are outside of the body connecting to the left and right rockers and to the rover body by a pivot in the center of the rover's top desk.**

(Page 22, lines 7-13) **All six driving wheels are independently driven, and only the front two wheels and rear two wheels have additional steering degree of freedom. There is a steering offset that is the distance between the steering axis and the front and rear wheels. Each wheel on Yutu-2 rover is equipped with an aluminum alloy hub attached with mesh and 24 grousers evenly arranged in two staggered rows, which is different from wheels on NASA's Mars rovers in design, as illustrated in Supplementary Fig. 10. In such a specific configuration, Yutu-2's wheels are light-weighted while having sufficient loading capability and great traction performance.**

64. Lindemann, R. A., Voorhees, C. J. Mars Exploration Rover mobility assembly design, test and performance. In *Proceedings of the IEEE International Conference on Systems, Man and Cybernetics*, Waikoloa, USA, 2005.

[redacted]

Response to Comment 8 (2). Compared with other mobility architectures listed in Supplementary Table S2 with examples, Yutu-2's rocker-bogie suspension enjoys the following advantages: 1) wheel's pressure on the ground is equilibrated, and the suspension system maintains a relatively constant weight on each wheel of the rover; 2) all six wheels nominally remain in contact with the surface when climbing over hard, uneven terrain, helping to propel the vehicle over the terrain; 3) the balance of the body is kept at an average angle of two sides rockers through differential mechanism, helping

to reduce the body's pitching and to provide a stable platform with the scientific instruments; 4) the structure absorbs significant energy from driving loads.

We have added the advantages of selecting rocker-bogie suspension as mobility architecture in the revision as follows. However, this paper is focused on the scientific exploration of the planetary rover, rather than the rover configuration design, thus we don't include the suspension configuration table of different planetary rovers into the paper.

(Pages 22, lines 3-7) Compared with other mobility architectures, the rocker-bogie suspension enjoys several advantages⁶⁴: 1) equilibrated ground pressure on wheels for consistent weight distribution; 2) continuous contact of all six wheels on rugged terrain, aiding propulsion.; 3) stabilized body angle with differential mechanism, providing a steady instrument platform; 4) absorbing driving load energy for enhanced structural durability.

64. Lindemann, R. A., Voorhees, C. J. Mars Exploration Rover mobility assembly design, test and performance. In Proceedings of the *IEEE International Conference on Systems, Man and Cybernetics*, Waikoloa, USA, 2005.

[redacted]

Comment 9. Supplementary Fig 3a - should be lateral and not lateral

Response to Comment 9. Thank you for the careful check. We have revised it and thoroughly gone through the entire paper and meticulously checked for any typos.

Overall, this is an excellent paper!

Response: We greatly appreciate the reviewer's encouraging comments.

REVIEWERS' COMMENTS

Reviewer #1 (Remarks to the Author):

The authors of the manuscript have thoroughly responded and addressed my suggested edits and I believe the manuscript is ready for publication.

Reviewer #2 (Remarks to the Author):

Thank you for addressing my comments. This paper is very informative. Looking forward to future missions!